# Gene expression profiling reveals insights into infant immunological and febrile responses to group B meningococcal vaccine

Daniel O'Connor[1,2,*] iD, Marta Valente Pinto[1,2], Dylan Sheerin[1,2] iD, Adriana Tomic[1,2,3], Ruth E Drury[1,2], Samuel Channon-Wells[1,2] iD, Ushma Galal[4], Christina Dold[1,2], Hannah Robinson[1,2], Simon Kerridge[1,2], Emma Plested[1,2], Harri Hughes[1,2], Lisa Stockdale[1,2], Manish Sadarangani[5], Matthew D Snape[1,2], Christine S Rollier[1,2] iD, Michael Levin[6] & Andrew J Pollard[1,2]

## Abstract

***Neisseria meningitidis*** **is a major cause of meningitis and septi-caemia. A MenB vaccine (4CMenB) was licensed by the European Medicines Agency in January 2013. Here we describe the blood transcriptome and proteome following infant immunisations with or without concomitant 4CMenB, to gain insight into the molecular mechanisms underlying post-vaccination reactogenicity and immunogenicity. Infants were randomised to receive control immunisations (PCV13 and DTaP-IPV-Hib) with or without 4CMenB at 2 and 4 months of age. Blood gene expression and plasma proteins were measured prior to, then 4 h, 24 h, 3 days or 7 days post-vaccination. 4CMenB vaccination was associated with increased expression of *ENTPD7* and increased concentrations of 4 plasma proteins: CRP, G-CSF, IL-1RA and IL-6. Post-vaccination fever was associated with increased expression of *SELL*, involved in neutrophil recruitment. A murine model dissecting the vaccine components found the concomitant regimen to be associated with increased gene perturbation compared with 4CMenB vaccine alone with enhancement of pathways such as interleukin-3, -5 and GM-CSF signalling. Finally, we present transcriptomic profiles predictive of immunological and febrile responses following 4CMenB vaccine.**

**Keywords** paediatrics; proteomics; systems biology; transcriptomics; vaccines
**Subject Categories** Immunology; Microbiology, Virology & Host Pathogen Interaction
**Mol Syst Biol. (2020) 16: e9888**

## Introduction

*Neisseria meningitidis* is estimated to cause 0.5–1.2 million infections and 50,000–135,000 deaths worldwide each year (Stephens *et al*, 2007; Rouphael & Stephens, 2012). Clinical outcomes of invasive infection vary by setting and strain, but even in resource-rich countries, permanent neurological sequelae are common in survivors and 5–10% of those affected die (Wilder-Smith & Memish, 2003). While susceptibility to invasive meningococcal disease is not completely understood, an inverse relationship is seen with the prevalence of complement-dependent serum bactericidal antibody (SBA) titres and the incidence of meningococcal meningitis (Goldschneider *et al*, 1969). Moreover, the levels of SBA correlate with post-immunisation protection from meningococcal disease at the population level (Andrews *et al*, 2003).

Capsular group B organisms cause most invasive meningococcal disease in developed countries (Pollard, 2004). In 2013, a multicomponent group B meningococcal vaccine (4CMenB, Bexsero, GSK), containing outer membrane vesicles from a meningococcal strain (NZ98/254) and three recombinant proteins, was licensed (Rollier *et al*, 2015). Subsequently, 12 European countries, as well as Canada, Australia and some Latin American countries, have issued recommendations related to its use in infancy (ECDC, 2017). In late 2015, the UK introduced this vaccine into a national, infant immunisation programme. A two-dose primary schedule of 4CMenB has been shown to be highly effective in preventing group B infection in infants (Parikh *et al*, 2016). However, when given concomitantly with other routine infant immunisations, fever ($\geq$ 38°C) is a common adverse reaction (> 60%) (Gossger *et al*, 2012; Martinón-Torres *et al*, 2017). While post-vaccination fever is generally a mild and benign event, it can concern parents and healthcare professionals and is indistinguishable from coincidental intercurrent infection,

1  Department of Paediatrics, University of Oxford, Oxford, UK
2  NIHR Oxford Biomedical Research Centre, Oxford University Hospitals NHS Foundation Trust, Oxford, UK
3  Institute of Immunity, Transplantation and Infection, Stanford University School of Medicine, Stanford, CA, USA
4  Nuffield Department of Primary Health Care, Clinical Trials Unit, University of Oxford, Oxford, UK
5  Department of Pediatrics, University of British Columbia, Vancouver, Canada
6  Division of Infectious Diseases, Department of Medicine, Imperial College London, London, UK
   *Corresponding author. Tel: +44 1865611365; E-mail: daniel.oconnor@paediatrics.ox.ac.uk

resulting in unnecessary medical visits, invasive procedures and laboratory investigations, which have an associated healthcare burden (Kapur *et al*, 2017; Nainani *et al*, 2017). Prophylactic paracetamol (acetaminophen) has been shown to reduce fever rates post-vaccination (Prymula *et al*, 2014). Therefore, various public health agencies (including Public Health England) have advised the use of prophylactic paracetamol for primary immunisations (at 2 and 4 months of age), when this vaccine was introduced into their respective infant immunisation programmes. Despite this recommendation, to which there appears good compliance (~ 94%), data from the United Kingdom have shown an increase in hospital attendances (estimated 1,440 additional visits per year) with transient adverse events following immunisation (AEFIs) since the introduction of the 4CMenB vaccine (Kapur *et al*, 2017; Murdoch *et al*, 2017; Nainani *et al*, 2017). Current clinical guidance for managing febrile infants is justifiably cautious, as it is often difficult to distinguish benign causes of fever (such as viral infections or vaccination) from the small proportion who have life-threatening bacterial infections (Craig *et al*, 2010; Esposito *et al*, 2014; Irwin *et al*, 2017). Consequently, post-vaccination hospital admissions are resulting in additional and unnecessary laboratory tests, invasive procedures (e.g. lumbar punctures) and/or antibiotic usage (Kapur *et al*, 2017). These are important factors when considering the implications associated with vaccine reactogenicity. Moreover, unfavourable vaccine reactogenicity may impact uptake of the specific vaccine or vaccines in general, with important implications. There is a need to understand both vaccine and vaccinee factors underlying vaccine reactogenicity, not least because this will guide the development of a new generation of vaccines with optimal reactogenicity profiles while preserving (or enhancing) immunogenicity. Recent reports have shown the power of peripheral blood RNA signatures in discerning the aetiology of acute febrile illness in children (Kaforou *et al*, 2013; Anderson *et al*, 2014; Herberg *et al*, 2016; Mahajan *et al*, 2016; Wright *et al*, 2018). Moreover, studies have also shown the fitness of such approaches in elucidating the molecular mechanisms underlying, and even predicting, immune responses, both in the context of vaccination and infectious challenge (Tsang *et al*, 2014; Blohmke *et al*, 2016; Davenport *et al*, 2016; Hemingway *et al*, 2017; von Both *et al*, 2018). Here we leveraged contemporary transcriptomic and proteomic approaches to dissect the mechanisms involved in both the reactogenicity and immunogenicity following infant immunisation. We then recapitulated these finding in a mouse model, further exploring the specific vaccine components underlying the early blood transcriptome following 4CMenB vaccination.

# Results

### Fever is more commonly observed in 4CMenB-vaccinated infants

One hundred and eighty-one (92 test and 89 control group) infants completed this study (Fig 1, Appendix Fig S1 and Dataset EV2). All infants had a baseline blood sample; and 28, 31, 30 and 36 infants had a blood sample at 4 h, 24 h, 3 days and 7 days, respectively. After 4CMenB plus control vaccination, the proportion of infants with at least one temperature recording $\geq 38°C$ within 24 h of receiving their 4-month dose of study vaccines was higher (59.6%

[95% CI: 48.6, 69.8]) than in infants who received the control vaccines alone (27.3% [95% CI: 18.3, 37.8]; Dataset EV3). Likewise, a statistically significant difference was observed in the Kaplan–Meier curves of time-to-fever event between the two vaccine study arms (log-rank test, $P < 0.001$; Fig 2A).

### Increase in neutrophils post-vaccination is associated with global blood transcriptional perturbation

Principal component (PC) analysis of blood transcriptome RNA-sequencing data from all study time points showed clustering of early (< 24 h) post-vaccination samples on the first and second PCs (Fig 2B). Analysis of the genes with the greatest contribution to this clustering revealed that *CD177*, a neutrophil-specific gene involved in neutrophil activation, was the leading contributor to (i.e. correlate with) PC 2 (Fig 2B). Full blood count (FBC) analysis showed neutrophil counts increased 4 h after vaccination (both test and control group), peaking at 24 h, before returning to baseline by 3 days (Appendix Fig S2). The neutrophil fraction determined from the transcriptome data (CIBERSORT) mirrored neutrophil counts in the FBC, and the two measures were highly correlated (rho = 0.81, $P < 2.2 \times 10^{-16}$; Fig 2C and D and Appendix Fig S2). PC analysis of CIBERSORT cell fraction data showed clustering of early post-vaccination samples, and the most important contributor to this clustering (PC1) was the neutrophil count (Appendix Fig S3). These analyses reveal that an increase in neutrophils early post-vaccination is associated with comprehensive changes to the underlying structure to the blood transcriptome (Fig 2B).

### Peak in blood transcriptome differential gene expression 24 h after infant vaccination

In this study, the greatest number of differentially expressed genes (DEGs, FDR < 0.01) compared with baseline (pre-vaccination at 4 months), when the two vaccine groups (test and control) were combined, was seen 24 h post-vaccination (DEGs = 5,553; Fig 3). It was noted that more upregulated genes surpassed the threshold for differential expression than downregulated genes (Fig 3A–D). However, this asymmetry was attenuated when transcriptomic data were corrected for neutrophil cell abundance (Appendix Fig S4). Of note, adjusting for neutrophil abundance reduced the number of DEGs at 4 h (719 vs. 161 DEGs) and 24 h (5,553 vs. 3,228 DEGs) post-vaccination (Appendix Fig S4). DEGs were also observed at the later time points, although fewer than at the early time points, 3 days (DEGs = 159) and 7 days (DEGs = 6; Fig 3C and D).

### Early gene regulation is generally consistent between vaccine groups

We next investigated whether gene regulation differed between infants who received the control vaccines only compared with those who received the additional test vaccine (4CMenB). Moreover, when all DEGs in either study vaccine regimen were considered, they generally showed similar regulation (i.e. same directionality; Appendix Figs S5 and S6). Correspondingly, blood transcriptional module analysis of the early study time points showed both the test and control groups upregulated genes associated with neutrophils and monocytes (Fig 3E). The immune activation module remained

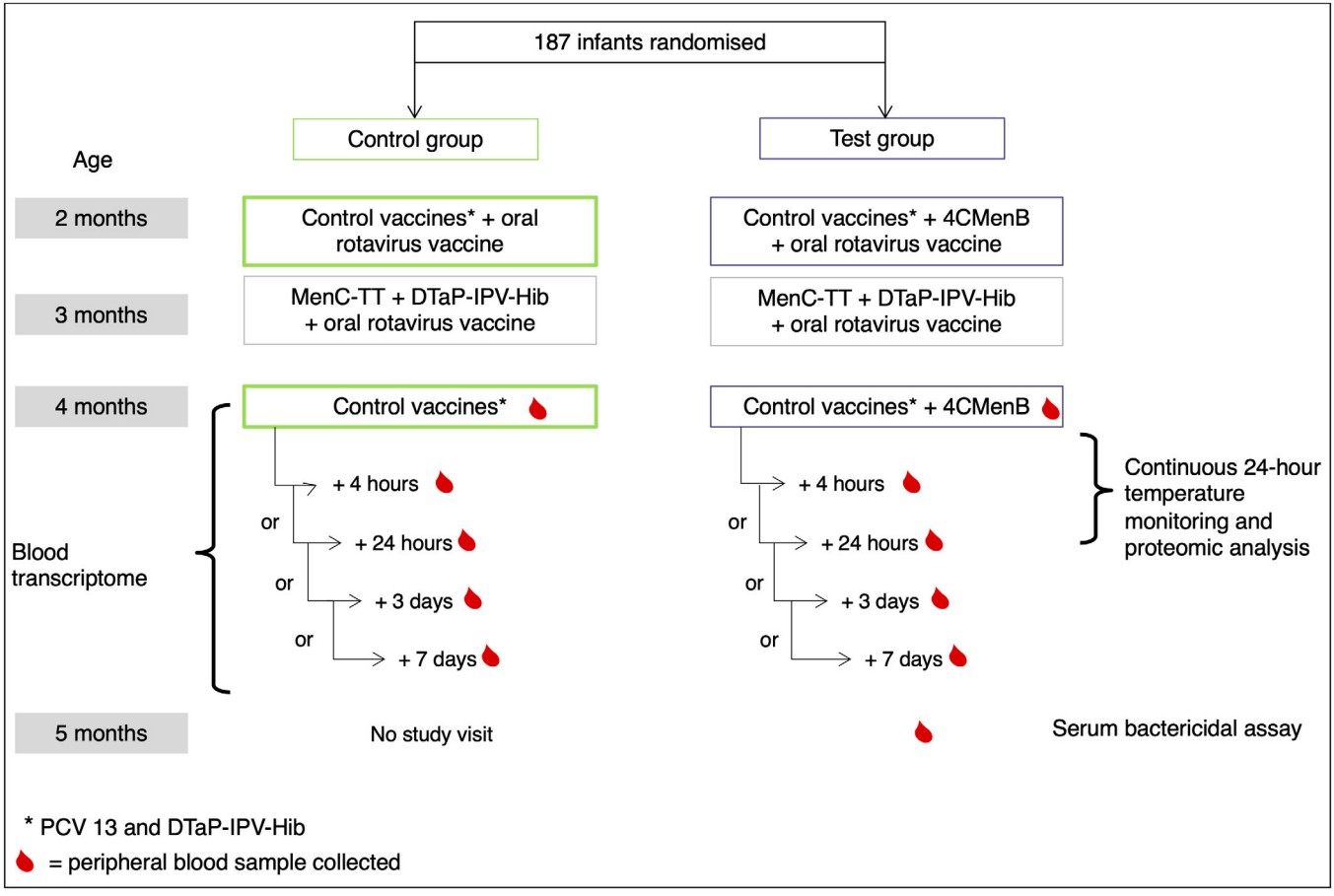

**Figure 1. Study overview**

enriched until 3 days post-vaccination, albeit with a smaller effect size than at 4- and 24-h time points (Fig 3E). At 7 days post-vaccination, DEGs were enriched for genes associated with the plasma cells gene module (Fig 3E). This similarity in early transcriptional signatures between the vaccine groups was also illustrated in gene ontology analysis (Appendix Figs S7–S9).

### Divergent transcriptomic and proteomic signatures in infants who received 4CMenB

While most transcriptional regulation was similar in both the study vaccine arms, three genes (*ENTPD7* [24 h], *IGHG3* [day 7] and *IGLV9-49* [day 7]) were significantly differentially regulated (FDR < 0.05) in infants who received 4CMenB compared with those who received the control vaccines alone (Fig 4A–D, Datasets EV4–EV7). The time course expression profiles of the top genes (ranked by *P*-value) differing between vaccine treatments are displayed in Fig 4.

A panel of 25 plasma proteins concentration was evaluated prior to vaccination and then at 4 and 24 h after vaccination. The concentration of five of these plasma proteins changed (FDR < 0.05) at 24 h post-vaccination (Dataset EV8). Moreover, the concentrations of four of these protein—CRP, G-CSF, IL-1RA and IL-6—were higher (FDR < 0.05) in the test group compared

with the control vaccine group, 24 h post-vaccination (Fig 5). These plasma cytokines were also highly correlated with the neutrophil fraction, as determined from the transcriptome data (CIBERSORTx; Appendix Fig S10).

### Identification of early gene signature associated with fever following infant vaccination

We next explored whether there were transcriptional differences between infants who experienced a fever within 24 h of vaccination and those who remained afebrile, irrespective of the which vaccines they received. We found gene regulation to be generally consistent between these subcategories of infants (Fig 6A and B). However, a single gene (*SELL*) was statistically significantly differentially regulated (FDR < 0.05) between febrile and afebrile infants, 4 h post-vaccination (Fig 6C and Dataset EV9).

### Baseline gene profiles predictive of immune response and reactogenicity

To explore the ability of baseline transcriptome data to predict subsequent vaccine-induced reactogenicity, we used a recently described approach that optimises a machine learning workflow through a Sequential Iterative Modelling "OverNight" (SIMON)

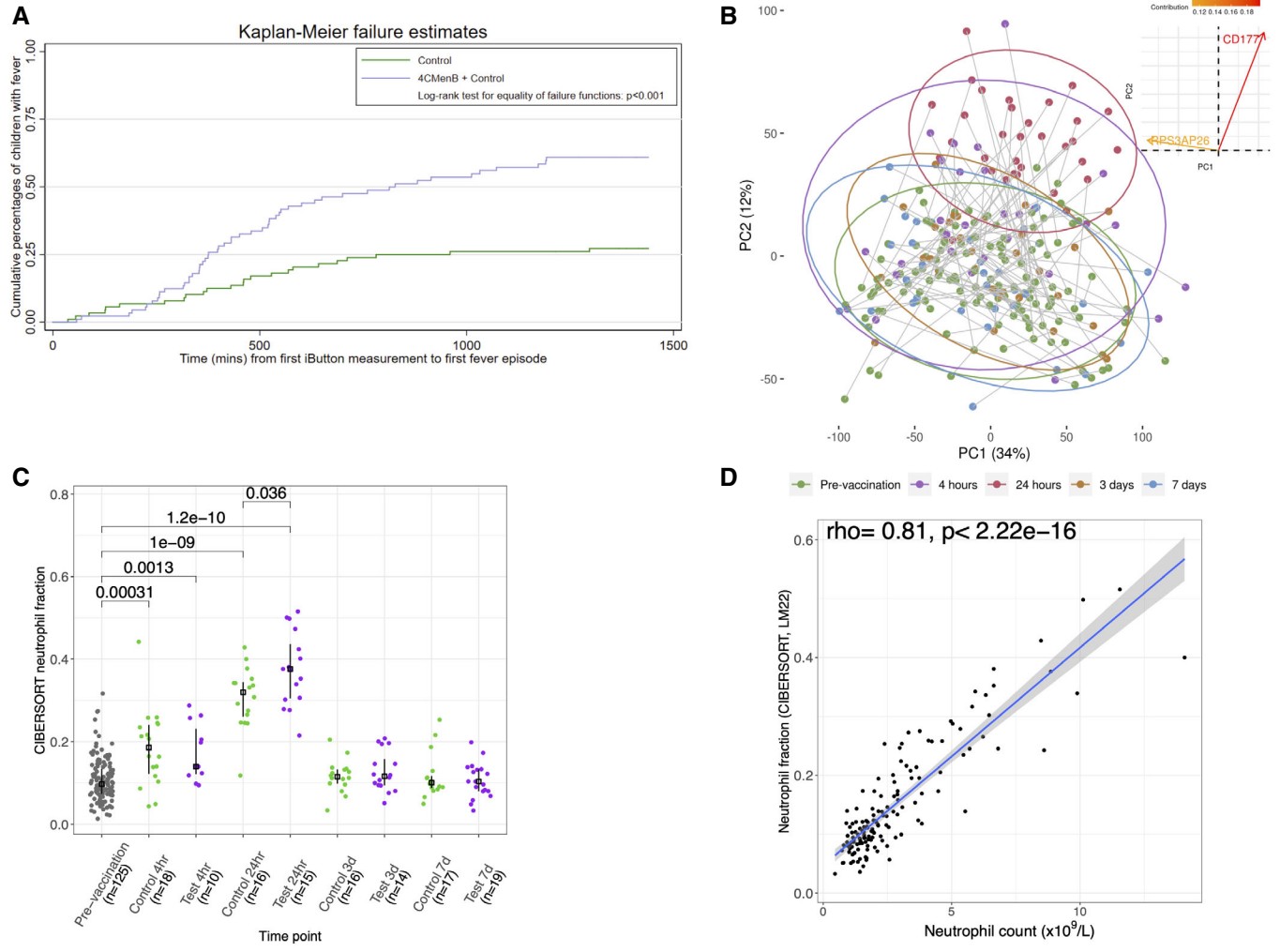

**Figure 2. Fever incidence within 24 h of study vaccines.**

A   Kaplan–Meier to first fever (≥ 38°C) episode within 24 h of receiving vaccines administered at 4 months of age; control $n = 88$ and 4CMenB $n = 89$.

B   Principal component (PC) analysis of RNA-sequencing data (14,837 genes, $n = 253$) from all study time points. The lines connect participant's pre- and post-vaccination samples. The ellipses are the two-dimensional (PC1 and PC2) 95% confidence intervals for each study time point. A contribution plot (top right) displays the genes contributing most to PC1/PC2, i.e. 0.15 implies 0.15% contribution of that variable to the principal components displayed.

C   Plotted are the CIBERSORTx neutrophil fractions from whole blood RNA-sequencing data, with median and interquartile range. *P*-values were determined from a two-sample Wilcoxon rank sum test. The number of individuals in each group is display in the x-axis.

D   Spearman's rank correlation between neutrophil counts measure by full blood counts ($n = 142$) and those estimated by CIBERSORTx, using the LM22 signature matrix.

(Tomic *et al*, 2019). SIMON, an automated machine learning approach, compares results from more than 100 different algorithms (Tomic *et al*, 2019). Here, we applied SIMON to baseline data before vaccination obtained from 54 infants that received 4CMenB vaccine in the test group. The outcome was determined as "fever" if the temperature recording was ≥ 38°C in the first 24 h post-vaccination. Since the number of genes is far greater than the number of donors, to avoid "curse of dimensionality", i.e. poor predictive power (Bellman, 1957), we performed feature selection before starting the SIMON analysis. The feature selection process facilitates the building a highly accurate models by focusing on the most important and relevant features (Bommert *et al*, 2020). To identify the most important features we utilised

two different approaches, (i) using DEGs in either febrile or afebrile infants at the two early study time points (i.e. 4- and 24-h post-vaccination) or (ii) PCA analysis, as reported previously (Golub *et al*, 1999; Song *et al*, 2010). In total, five different datasets were generated (as described in Methods) for SIMON analysis. Models built with SIMON on the datasets generated using DEGs had a maximum AUROC of 0.68 and 0.75, for DEGs at the 4- and 24-h time points, respectively (Datasets EV10 and EV11). Automated model selection algorithms perform well on features selected using the PCA, as previously described (Kohavi & John, 1997). We performed SIMON analysis on datasets containing the top 500, 200 and 100 transcripts contributing to PC1 (Datasets EV13–EV15). The top performing models were

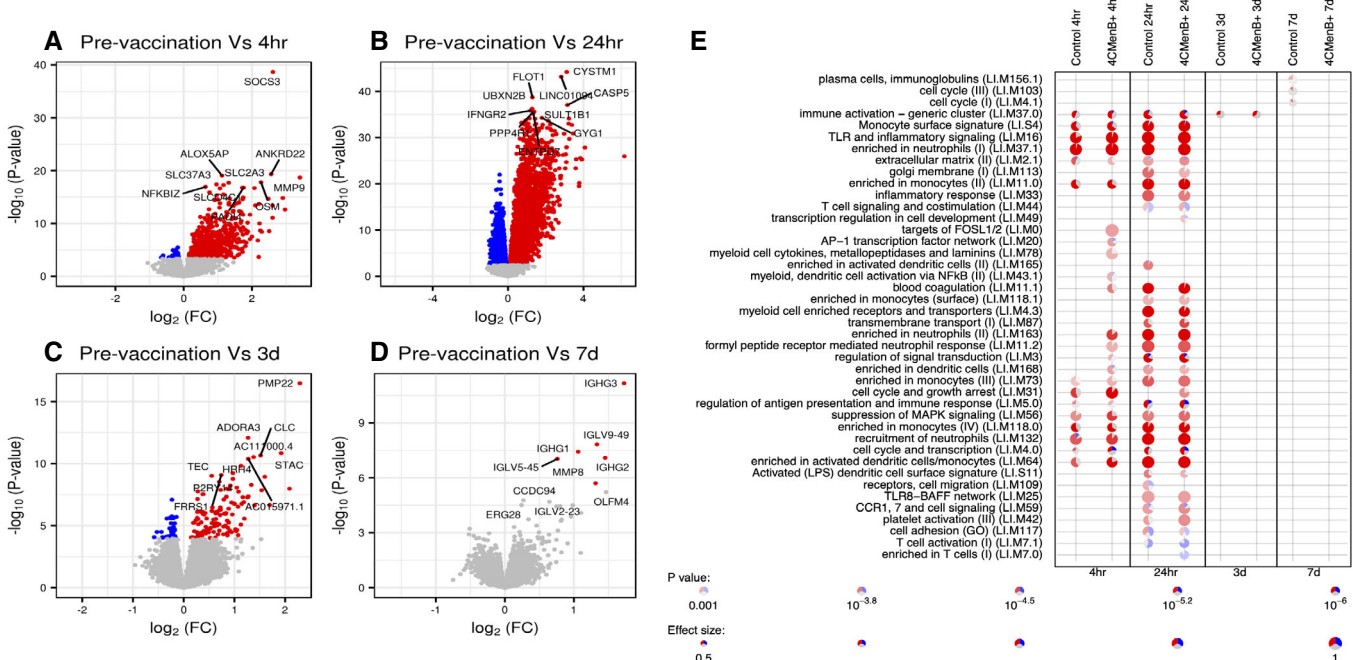

**Figure 3. Blood gene signatures following infant vaccination.**

A  Volcano plot highlighting differentially expressed genes (DEGs, false discovery rate [FDR] < 0.01; red upregulated and blue downregulated) at each study time point versus pre-vaccination (4 months of age) 4 h post-vaccination (719 DEGs, *n* = 28). *P*-values were obtained from the moderated t-statistic, after adjustment for multiple testing (Benjamini and Hochberg's method). The top 10 genes, ranked by FDR, are labelled.

B  Same as (A) but 24 h post-vaccination (5,553 DEGs, *n* = 31).

C  Same as (A) but 3 days post-vaccination (159 DEGs, *n* = 30).

D  Same as (A) but 7 days post-vaccination (6 DEGs, *n* = 36).

E  Modular signature induced following infant vaccination. Enriched modules (FDR < 0.001) are displayed. Segments of the pie charts represent the proportion of upregulated (red) and downregulated (blue) genes (absolute fold change > 1.25). Enrichment *P*-values were derived from a hypergeometric test, after adjustment for multiple testing (Benjamini and Hochberg's method).

selected using the dataset containing 200 genes from the PC1 (39.8% variance explained by the PC1; Appendix Fig S12A and Dataset EV12).

Out of 215 machine learning algorithms evaluated, SIMON successfully built seven models with good discriminative ability calculated as area under ROC (AUROC) > 0.7 on both training and withheld test set (Dataset EV13). The model with the highest performance measurements was built using a sparse distance weighted discrimination algorithm (Wang & Zou, 2016) with train AUROC of 0.7867 and test AUROC of 0.7 (Fig 6D and Appendix Fig S12B). Of the total 200 genes, three genes, *APBA3*, *AASS* and *FKBP4*, were able to discriminate between infants that developed fever and those who remained afebrile (Fig 6E and Dataset EV16). *APBA3* had the highest contribution with variable importance score 100 in three other models also built with good predictive measurements (Fig 6D and E). Moreover, lower performance was observed (train AUROC 0.6) with the removal of *APBA3*, as evaluated using the dataset that contained only 100 transcripts from the PC1 (i.e. no *APBA3* included; Dataset EV14). Infants that developed fever after 4CMenB vaccine had lower expression levels of *APBA3* and *FKBP4* and higher expression levels of *AASS* transcripts in their blood before vaccination compared with infants who remained afebrile (Fig 6F). We also explored the ability of baseline transcriptomic data to predict subsequent vaccine-induced

immunity. The 200 transcripts, that contributed the most to the variance explained by the first PC, were used to train a SVR model to predict post-vaccination MenB SBA titres (Fig 6G and Dataset EV12). The top 5 genes contributing to this model are shown in Fig 6I. This model was shown to predict SBA titres that correlated (*r* = 0.82, *P* = 0.007) with observed MenB-specific SBA titres in an test set of individuals (not used to train model; Fig 6H). We did not observe a difference in post-vaccination MenB-specific SBA titres between infants who experienced a fever within 24 h of vaccination and those who remained afebrile (Appendix Fig S11B).

**4CMenB induces a greater magnitude of early pro-inflammatory gene expression in mice when administered in combination with control immunisations**

In order to validate genes significantly differentially regulated 24 h post-immunisation in infants, we proceeded to immunise mice with control vaccines, on their own or in combination with 4CMenB (test group), or with 4CMenB only and performed RNA-sequencing on RNA derived from the peripheral blood taken at 24 h after the second dose. Mice in the test group exhibited greater perturbation of the transcriptome (695 DEGs vs PBS controls), compared with those receiving the 4CMenB vaccine alone (268 DEGs vs PBS controls) or

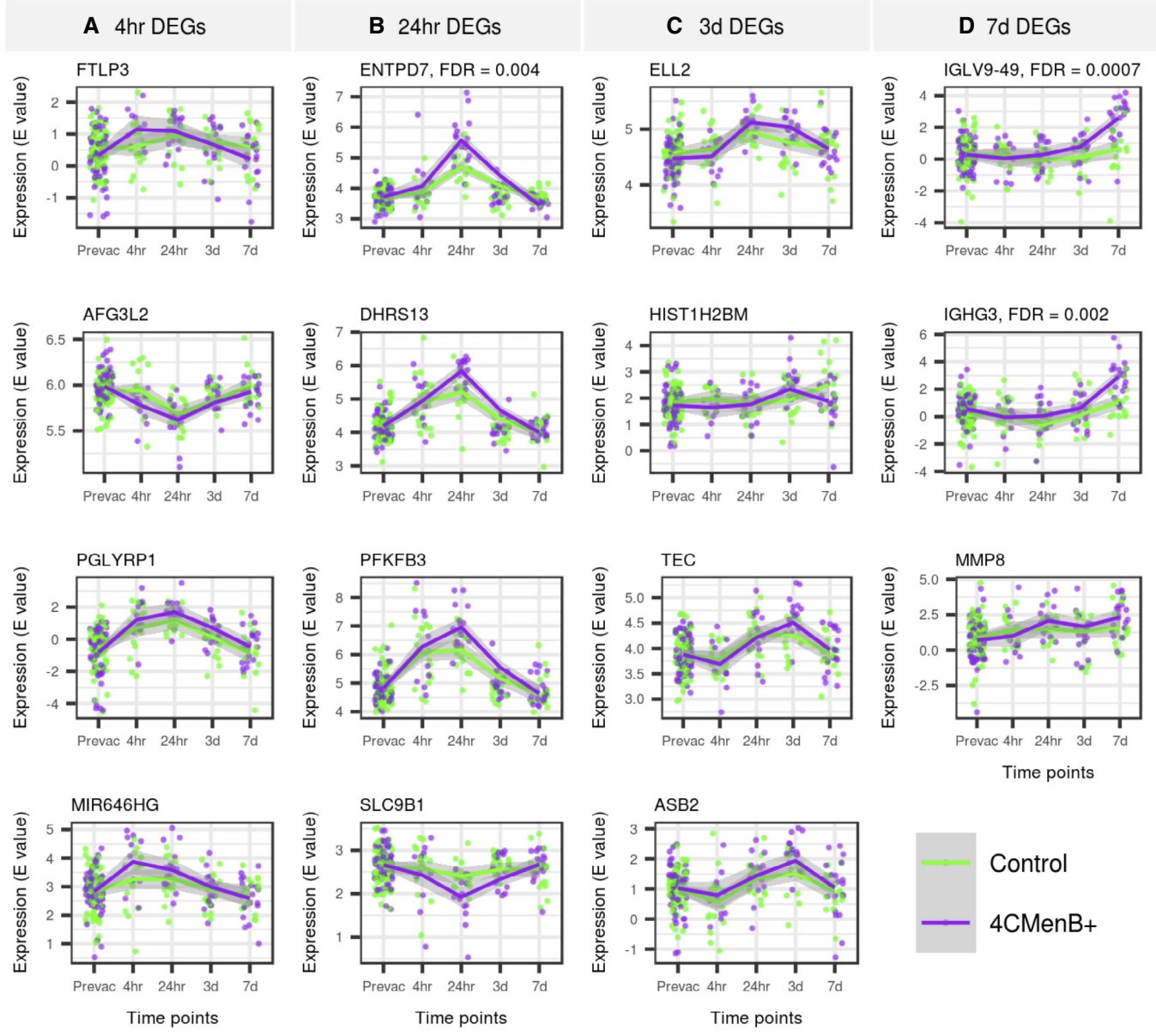

**Figure 4. Differential gene regulation between vaccine groups.**

A Top genes differentially regulated between vaccine groups 4 h post-vaccination. Note, only genes differentially expressed in either study group (concomitant 4CMenB or control vaccines alone) were included in intergroup analysis. The plotted lines are the LOESS (locally estimated scatterplot smoothing) regression curves with the 95% confidence intervals in grey. The FDR was derived by comparing fold changes in gene expression in the control group with the test group, from baseline to the time point designated, and is reported if statistically significant (FDR < 0.05). *P*-values were obtained from the moderated t-statistic, after adjustment for multiple testing (Benjamini and Hochberg's method). Pre-vaccination samples *n* = 125, 4 h samples *n* = 28, 24 h samples *n* = 31, 3 day sample *n* = 30, 7 day samples *n* = 36. The expression E value is the gene expression value derived from the voom-limma workflow (Law *et al*, 2014).

B Same as (A) but top genes differentially regulated between vaccine groups 24 h post-vaccination.

C Same as (A) but top genes differentially regulated between vaccine groups 3 days post-vaccination.

D Same as (A) but top genes differentially regulated between vaccine groups 7 days post-vaccination.

control vaccines alone (one DEG vs PBS controls; Appendix Fig S13A).

Several genes encoding pattern recognition receptors (PRRs) and pro-inflammatory signalling pathway components were differentially regulated between the 4CMenB-containing combinations and the control vaccines (Fig 7). Pathways analysis disclosed significantly enriched gene sets in all vaccine groups including IL-1 signalling and toll-like receptor signalling (Appendix Fig S13C). Pathway analysis revealed that the test group was associated with enrichment of several immune pathways including IL-3, IL-5 and GM-CSF signalling, compared with the 4CMenB only group (Appendix Fig S13A).

Cell deconvolution analysis was performed with these samples to infer the composition of immune cells from whole blood gene signatures (Appendix Fig S14). A decrease in the proportion of most B-cell subsets was observed in the test and 4CMenB only groups. Neutrophils were increased in both 4CMenB-immunised groups but not the control immunisation group (Appendix Fig S15).

**Bacterial outer membrane components LPS and peptidoglycan recapitulate pyrogenicity profile evoked by 4CMenB vaccine**

Lipopolysaccharide (LPS), contained within the OMV component of the 4CMenB vaccine, has been identified as a potential source of vaccine reactogenicity (Dowling *et al*, 2016). However, other

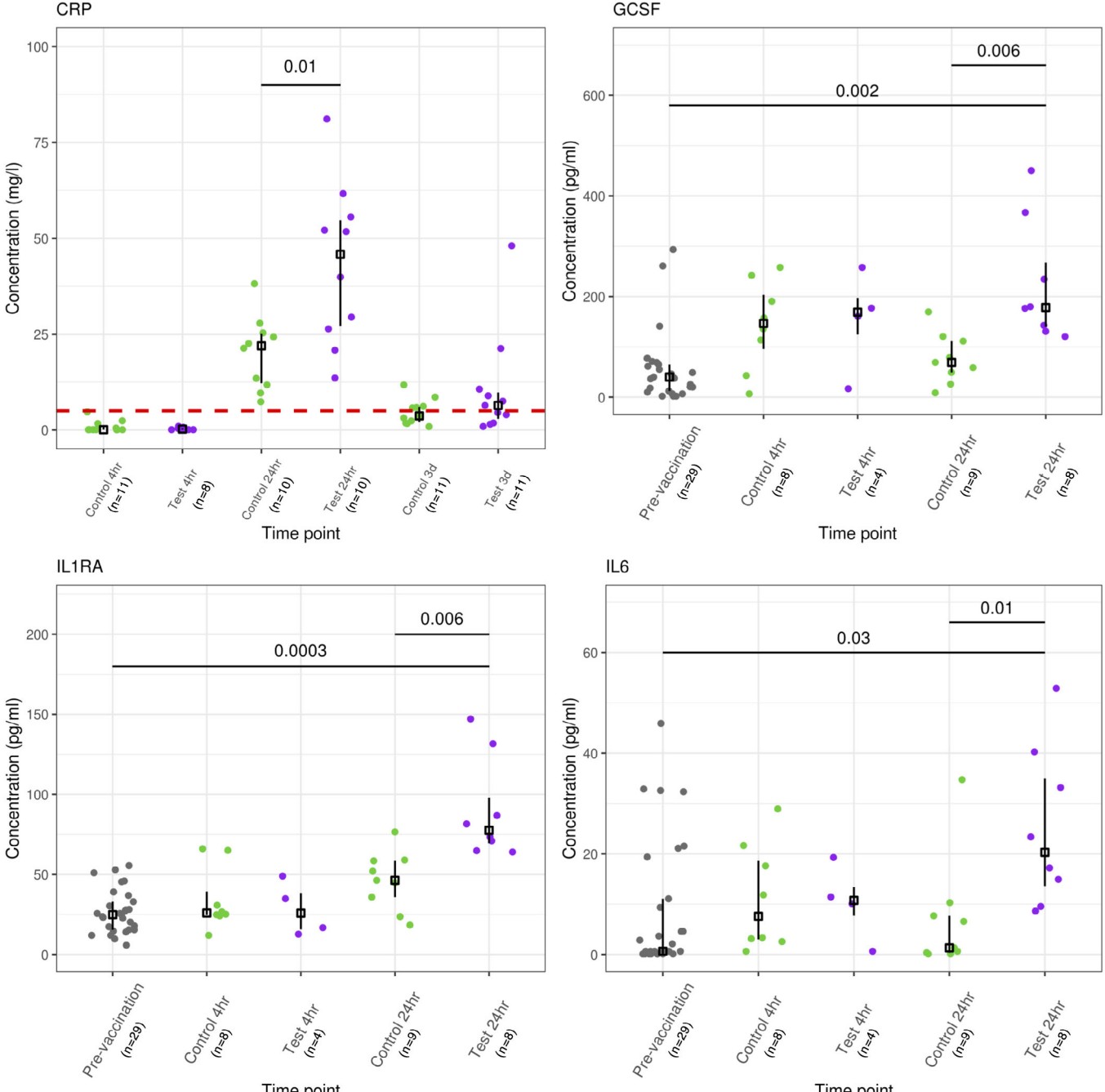

**Figure 5. Plasma proteins that differed in concentrations following vaccination in the test compare with the control vaccine group.**

False discovery rate (FDR) is displayed above and were derived from a two-sample Wilcoxon rank sum test (Benjamini–Hochberg correction). Vertical lines represent the interquartile range around the median. The red dashed horizontal line is upper limit of the normal CRP reference interval (0–5 mg/l). The number of individuals in each group is display in the x-axis.

components of the OMVs may also be pyrogenic, for example, naturally occurring OMVs contain peptidoglycan (van der Pol *et al*, 2015). As we observed upregulation of *Tlr2*, *Trem1* and *Pglyrp1*, known receptors for peptidoglycan, we assessed the temperature profiles of mice following immunisation with LPS, peptidoglycan or LPS + peptidoglycan (Fig 8). All LPS and peptidoglycan preparations evoked significant rises in temperature, with no rise in temperature observed in alum only, or unvaccinated, mice (Fig 8). Temperature rises following peptidoglycan were transient, but the highest temperatures were observed in the peptidoglycan + LPS group (Fig 8).

# Discussion

Here we describe extensive blood transcriptional perturbation following infant vaccination. We reveal a prominent change to the blood transcriptome from 4 h that peaks at 24 h post-vaccination, which is associated with a transient increase in peripheral blood neutrophils. We used a continuous temperature measurement device to describe significantly higher rates of fever in infants following 4CMenB plus control vaccines compared with control vaccines only. Furthermore, we detailed differences at both the transcriptomic and proteomic level, in infants who received the concomitant 4CMenB vaccine. Moreover, we showed enhanced early expression of a gene encoding a protein involved in leucocyte extravasation that was associated with fever following vaccination. Finally, we defined baseline whole blood gene signatures that were predictive of vaccine immunogenicity and reactogenicity.

There are an increasing number of studies describing transcriptional responses to immunisation in adults (Obermoser *et al*, 2013; Nakaya *et al*, 2015; O'Connor *et al*, 2017). However, there are few data describing transcriptional regulation following infant immunisation—despite most vaccination programmes being targeted to this population (ECDC, 2020; WHO, 2018). Here we present the first whole-transcriptome description of infant blood samples taken following immunisation. Consistent with previous studies in older children and adults, we observed an initial upregulation of genes involved in innate recognition of microbial motifs and the downregulation of genes associated with T cells (Fuller *et al*, 2007; Obermoser *et al*, 2013; Nakaya *et al*, 2016). We also revealed that much of the blood transcriptional perturbation occurring within 24 h of infant vaccination was attributable to post-vaccination neutrophilia. While neutrophils are principally described as short-lived effector cells, they also express numerous innate recognition receptors and produce a vast array of immune mediators and it has even recently been shown that they can be induced to present antigens to T cells (Mantovani *et al*, 2011; Vono *et al*, 2017). The role of this transient increases in circulating neutrophils on the regulation of vaccine responses is unclear; however, neutrophils migrate rapidly to the site of vaccination and are the first to transport antigen to afferent lymph nodes (Calabro *et al*, 2011).

Consistent with previous studies, fever ($\geq 38°C$) was more common in infants vaccinated with 4CMenB, given concomitantly with control (PCV13 + DTaP-IPV-Hib) immunisations, than in infants who received control immunisations alone (Gossger *et al*, 2012). Fever has an important role in fighting off infection (Evans *et al*, 2015). Furthermore, this rise in body temperature has been

implicated in a positive feedback loop during the early stages of inflammatory responses (Evans *et al*, 2015). Fever-range hyperthermia in mice increases circulating neutrophil counts and promotes extravasation of leucocytes via L-selection (Wang *et al*, 1998; Capitano *et al*, 2012). Here, we show the gene encoding L-selectin (*SELL*) is upregulated, 4 h after vaccination, in peripheral blood from infants who experience fever within 24 h of vaccination compared with those who remained afebrile. The upregulation of *SELL* suggests that post-vaccination fever may be associated with enhanced leucocyte extravasation and neutrophil recruitment to the site of inflammation (i.e. vaccination site) (Zarbock & Ley, 2008). There are data suggesting post-vaccination fever can be associated with increased immunogenicity (Andrews *et al*, 2011; Li-Kim-Moy *et al*, 2018). Correspondingly, prophylactic use of antipyretics has been associated with reduced vaccine immunogenicity, albeit the clinical relevance of this effect is debatable (Prymula *et al*, 2009; Prymula *et al*, 2014; Li-Kim-Moy *et al*, 2018). Moreover, as antipyretics (such as acetaminophen) have anti-inflammatory, as well as antipyretic effects, the mechanisms underlying their relationship with vaccine immunogenicity are unclear (Prymula *et al*, 2014; Saleh *et al*, 2016).

Next, we explored why concomitant 4CMenB is associated with increased reactogenicity by comparing the blood transcriptome profiles of infants who received this vaccine with those who received control vaccines alone. While the transcriptional changes induced by the two vaccine regimens evaluated were broadly similar, three genes showed statistically significant differences (FDR < 0.05) in their regulation: *ENTPD7* (24 h), *IGHG3* (day 7) and *IGLV9-49* (day 7). The first of these, *ENTPD7*, was significantly upregulated 24 h post-vaccination in infants who received 4CMenB compared with infants who received the control vaccines alone. *ENTPD7* encodes an ectonucleotidase, which is an ATP hydrolysing enzyme. Extracellular ATP modulates multiple immune cell functions, via purinergic receptors, and is tightly regulated by ectonucleotidases (Chen *et al*, 2006). Concurrently, plasma levels of G-CSF, the prototypical cytokine involved in the production and mobilisation of neutrophils, were elevated in those who received concomitant 4CMenB compared with the control vaccine alone (Bendall & Bradstock, 2014). These data support differences in neutrophil motility and chemotaxis between these vaccine groups.

We observed a raised CRP 24 h post-vaccination in both the test (median 45.84 mg/l; IQR: 27.17–54.72) and control (median 22.00 mg/l; IQR: 12.21–25.11), compared with the normal reference range for this plasma protein (0–5 mg/l). Moreover, CRP levels were higher in the test group than the control group, 24 h post-vaccination (FDR = 0.01). Importantly, an elevated CRP (> 5mg/l) was common 24 h post-vaccination—CRP levels and neutrophil counts are commonly used as diagnostic markers in suspected sepsis—these findings need to be taken into consideration when assessing febrile infants following vaccination (Van den Bruel *et al*, 2011; Faix, 2013). These data are consistent with a recent report showing increased CRP levels from 12 to 72 h post-vaccination, in premature infants administered 4CMenB with routine vaccines (Kent *et al*, 2019).

In addition to G-CSF and CRP, plasma protein levels of IL-6 and IL-1RA were also elevated in the test vaccine group compared with the control vaccine group. IL-6 is the archetypal member of the IL-6 cytokine family, with a central role in inflammation and immunity

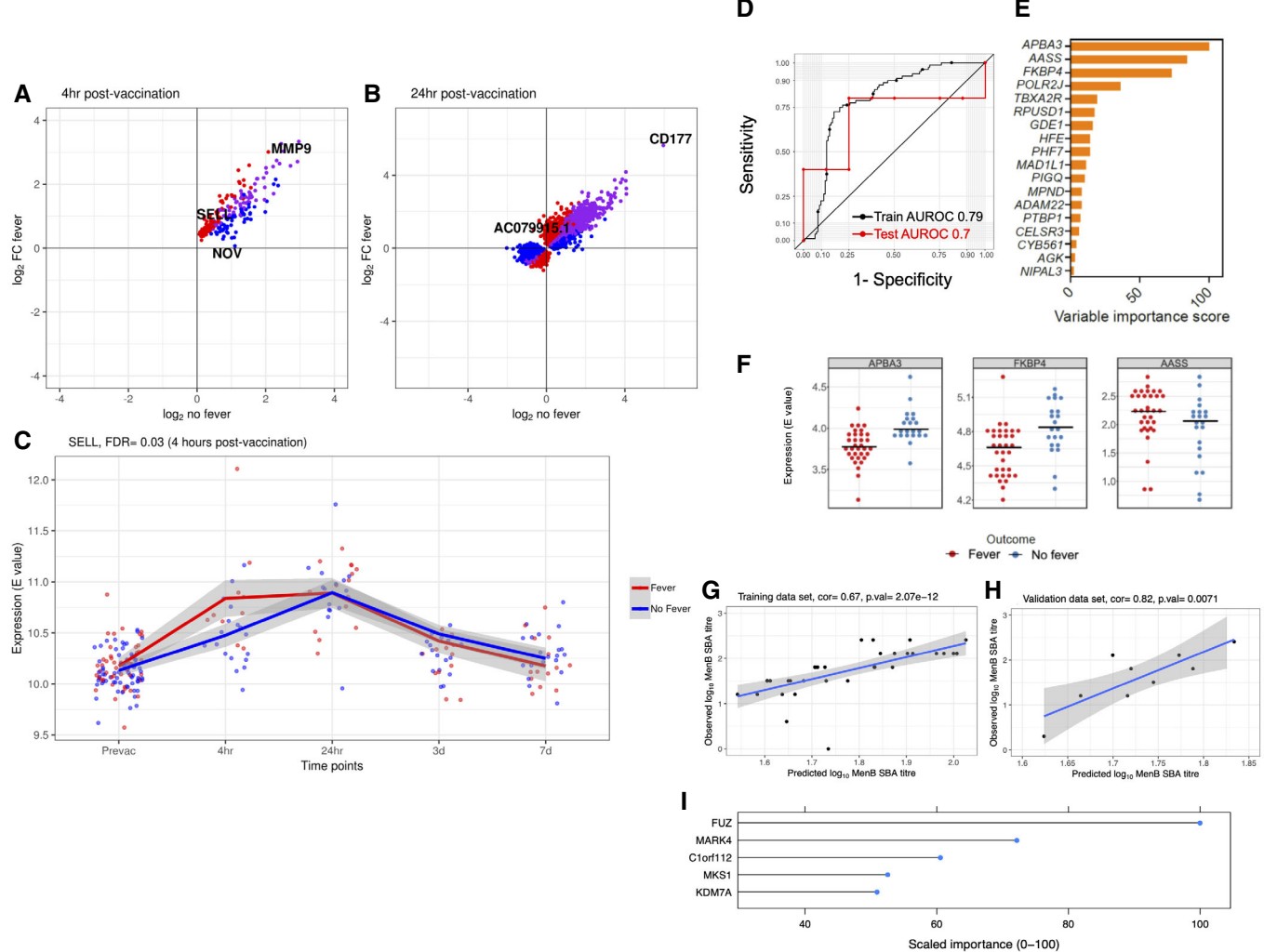

**Figure 6.  Comparison of blood gene profiles of infants who experienced post-vaccination fever with those who remained afebrile.**

A, B  Agreement plot. Red = differentially expressed (DE) in febrile infants, blue = DE in afebrile infants, purple = DE in febrile and afebrile infants.

C  Exemplar genes differentially expressed at 4 and 24 h, respectively, between infants who experienced a fever within 24 h of vaccination and afebrile infants. The plotted lines are the LOESS (locally estimated scatterplot smoothing) regression curves with the 95% confidence intervals in grey. The false discovery rate (FDR) was derived by comparing fold changes in gene expression between infants who experienced post-vaccination fever and those who remained afebrile. *P*-values were obtained from the moderated t-statistic, after adjustment for multiple testing (Benjamini and Hochberg's method). Pre-vaccination samples $n = 125$, 4 h samples $n = 28$, 24 h samples $n = 31$, 3 day sample $n = 30$, 7 day samples $n = 36$.

D  Performance of predictive model built with sparse distance weighted discrimination (sdwd) algorithm to predict fever following concomitant 4CMenB vaccine.

E  Variable importance score of the features from the sdwd model.

F  Expression levels (Limma E value) of the top three transcripts from the sdwd model in the individuals that develop fever ($n = 33$, red circles) and those that don't develop fever ($n = 21$, blue circles).

G–I  Support vector regression (SVR) performance of model to predict post-vaccination MenB-specific SBA titres, (G) training dataset ($n = 36$), (H) performance of model in the test dataset ($n = 9$), I) the top five genes ranked by importance from the SVR model.

(Jones & Jenkins, 2018). IL-6 can be produced by local sites of inflammation, travel through the blood to the liver where it induces a range of acute phase proteins, such as CRP (Heinrich *et al*, 1990). Conversely, IL-1RA is an IL-1R antagonist, so it is conceivable that this is acting to limit inflammation through some negative feedback loop (Garlanda *et al*, 2013).

We aimed to validate the infant gene expression data using a murine model. We immunised mice with control immunisations, with or without the 4CMenB test vaccine, and included a 4CMenB

only group. Firstly, we showed a greater magnitude of change in the blood gene profiles of mice immunised with 4CMenB vaccine than those immunised with control immunisations only. Moreover, cell deconvolution analysis showed mice immunised with 4CMenB also had an increase in their neutrophil gene signature, which was not observed in mice immunised with control immunisations only. Notably, the peptidoglycan receptor *Pglyrp1* and the triggering receptor expressed on myeloid cells *Trem1* were amongst the most upregulated genes in mice immunised with 4CMenB. PGLYRP1 has

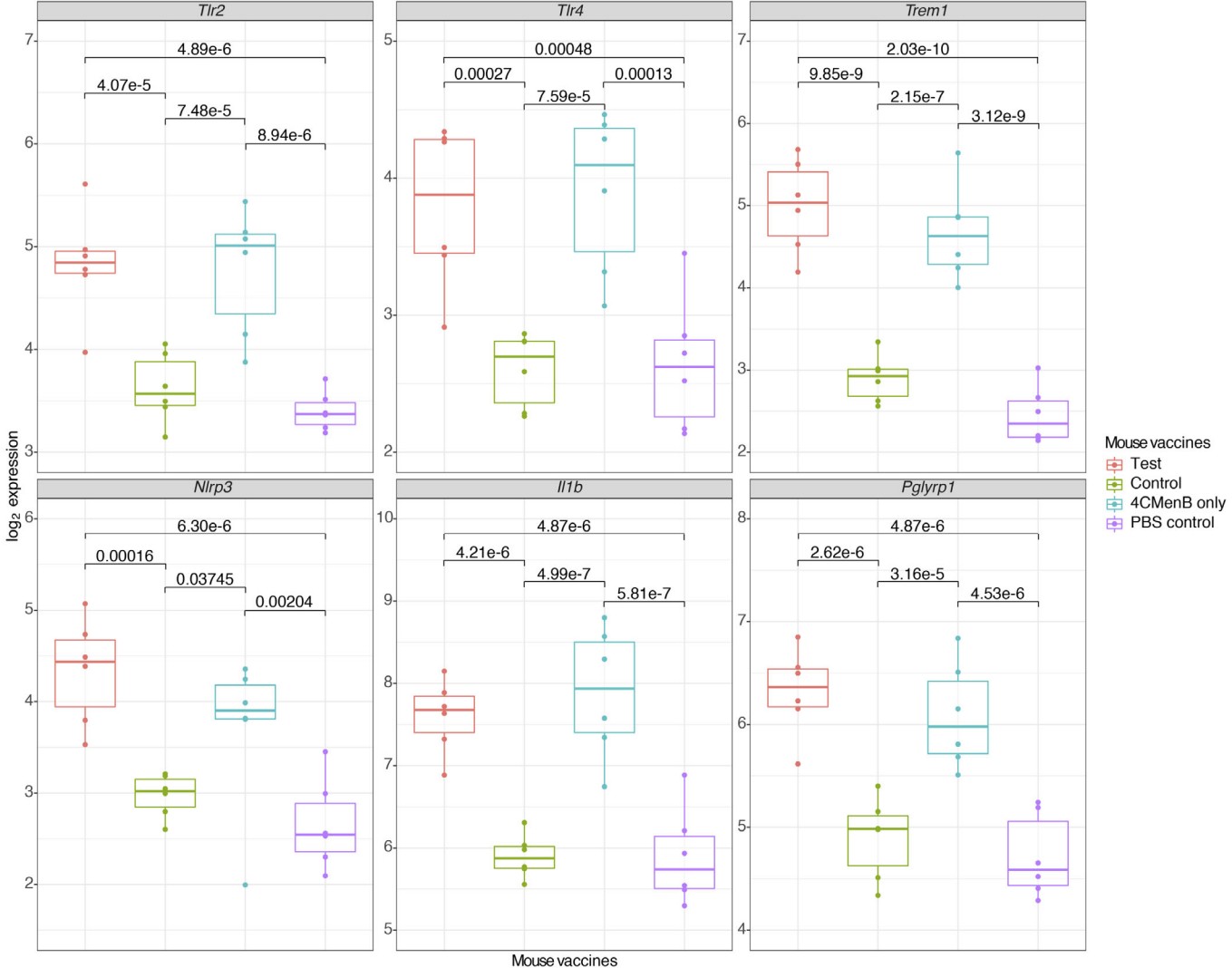

**Figure 7.  Selection of top differentially expressed genes identified in the mouse study.**
Boxplots represent the median with interquartile ranges. False discovery rate (FDR) is displayed above and was derived from a two-sample Wilcoxon rank sum test (Benjamini–Hochberg correction). N = 6 mice per vaccine group.

been characterised as a ligand for the TREM-1 receptor (Pelham *et al*, 2014). TREM-1 is upregulated on neutrophils and monocytes during bacterial infection and, when ligated, acts synergistically with LPS to amplify the pro-inflammatory response (Bouchon *et al*, 2001). In order for PGLYRP1 to stimulate TREM-1, it must be cross-linked with peptidoglycan, whereby it induces cytokine production in neutrophils and macrophages (Read *et al*, 2015).

In the infant study, we found *PGLYRP1* and *TREM1* to be significantly upregulated 4 and 24 h post-vaccination in both vaccine groups. Of note, 4 h after vaccination, *PGLYRP1* was one of the most differentially regulated genes in the concomitant 4CMenB group compared with the control vaccine group. Although *PGLYRP1* expression is restricted to neutrophils (in blood cells), correcting for neutrophil counts did not remove the difference in expression seen between the concomitant 4CMenB group and the control group (Liu *et al*, 2000). *PGLYRP1* expression is upregulated by stimulation with

peptidoglycan (a cell wall component of both Gram-positive and Gram-negative bacteria) (Uehara *et al*, 2005). Peptidoglycan fragments are known to covalently attach to some of the purified pneumococcal polysaccharides within PCV13 (Sørensen *et al*, 1990). Moreover, while the peptidoglycan content of detergent extracted OMVs (component of 4CMenB) has not been described, naturally occurring OMVs do contain peptidoglycan (van der Pol *et al*, 2015). One explanation for the differences seen in *PGLYRP1* regulation between those who received concomitant 4CMenB and those who received control vaccines alone is a dose-dependent difference in the amount of peptidoglycan delivered in the study vaccine regimens. Here, in mice, we showed that peptidoglycan alone could evoke a rise in temperature and presented data suggesting an additive pyrogenic effect of administration of LPS plus peptidoglycan. These data suggest that the increased reactogenicity of 4CMenB when given concomitantly with other immunisations, compared

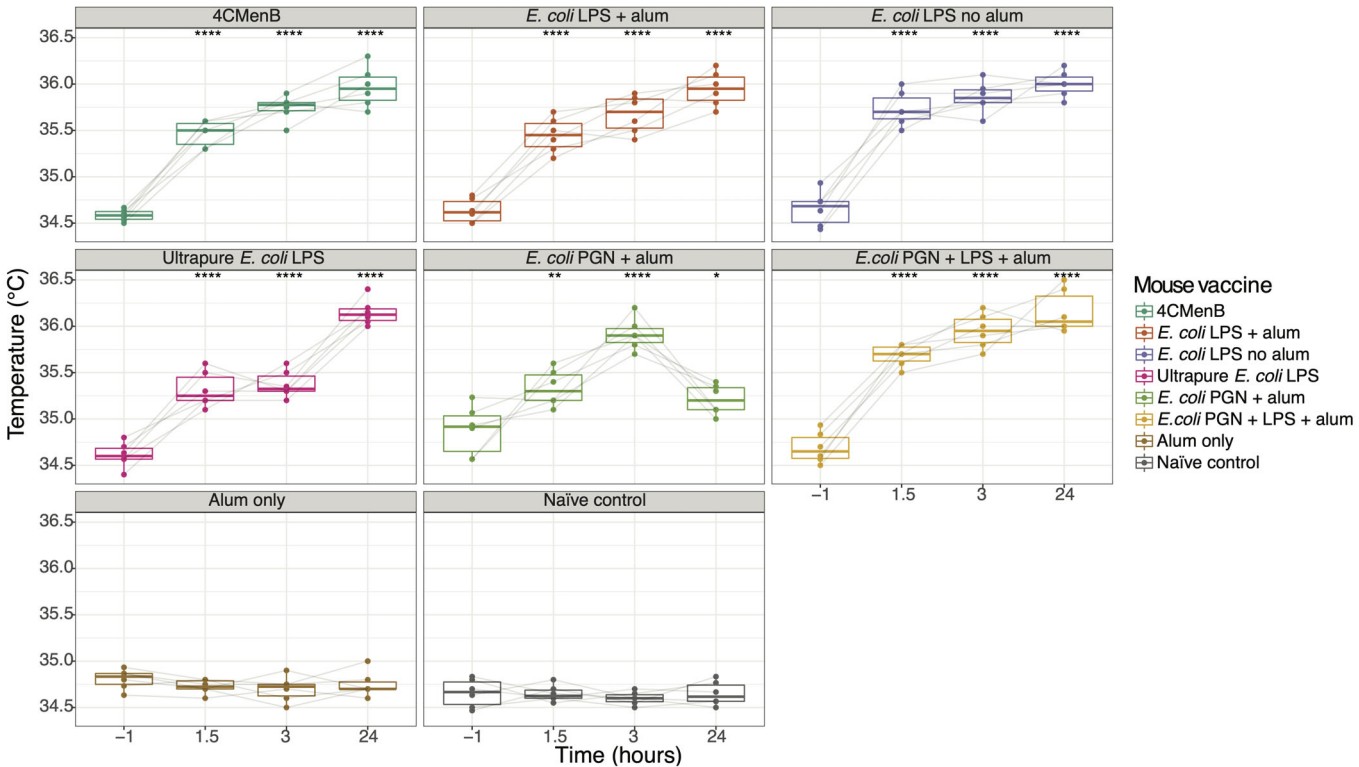

**Figure 8. Mouse temperatures following administration of purified preparations of Gram-negative bacteria outer membrane components.**

Boxplots depicting the change in temperature from baseline to 24 h after the second 1 µg dose of each of a variety of *Escherichia coli* lipopolysaccharide (LPS) and peptidoglycan (PGN) preparations, with and without alum, and combinations compared with 1/5 of the human dose of 4CMenB. Boxplots represent the median with interquartile ranges. Vaccine group medians at each time point were compared with baseline for that vaccine using a Wilcoxon signed-rank test. * < 0.05, ** < 0.01, **** < 0.0001. *N* = 6 mice per vaccine group.

with 4CMenB alone (Gossger *et al*, 2012) may be attributable to the additive effects of pyrogenic components of these vaccines.

A noteworthy feature of infant 4CMenB vaccination is that when administered alone, fever rates are similar to those seen after other infant immunisation regimens (PCV7 and DTaP-IPV-Hib) but when given concomitantly with other childhood vaccines, fever rates are increased (Gossger *et al*, 2012). This additional reactogenicity has been attributed to the OMV component, as concomitant vaccination with the recombinant 4CMenB proteins and other routine vaccines (DTaP-HBV-IPV/Hib and PCV7) displays lower reactogenicity than concomitant OMV-containing 4CMenB (Esposito *et al*, 2014). It has been suggested that attenuating LPS, by genetic modification (e.g. *lpxL1*), may ameliorate the reactogenicity of the OMV-containing vaccines (Dowling *et al*, 2016). However, *IL1β* (a pro-inflammatory cytokine and endogenous pyrogen) has been found to be upregulated similarly in mice immunised with native and *lpxL1* mutant OMV-containing vaccines, suggesting that non-endotoxin pyrogens, such as peptidoglycan, may also contribute to OMV reactogenicity (Sheerin *et al*, 2019). Moreover, in mice, the concomitant regimen was associated with increased gene perturbation compared with 4CMenB vaccine administered alone and enhancement of pathways such as interleukin-3, -5 and GM-CSF signalling. These findings highlight the importance of fully characterising the pyrogen content of vaccines and understanding how these may interact when

administered concomitantly. Despite substantial differences in peripheral blood neutrophil abundance, between mice and human infants (female 8-week C57BL/6 mice, neutrophil count ~ 0.25 × 10$^9$/l; 2- to 6-month human infants, neutrophil count 1–8.5 × 10$^9$/l), we observed similar trends in gene expression profiles between the species (Virgo, 2020; The Jackson Laboratory, 2007). While the mouse model has clear limitations, these data support the utility of this model in exploring experimental procedures not amenable in human subjects.

Consistent with previous reports, we observed the upregulation of genes associated with terminally differentiated, antibody-secreting B cells (plasma cells) at 7 days post-vaccination (Obermoser *et al*, 2013; O'Connor *et al*, 2017). Of note, we identified a particular gene encoding an immunoglobulin light chain molecule, *IGLV9-49,* that was upregulated after vaccination with 4CMenB vaccine compared with control vaccines alone. The immunoglobulin heavy chain is often thought to dominate antigen recognition (Xu & Davis, 2000). However, the prominence of the light chain in determining the specificity and affinity of antigen binding has also been demonstrated (Song *et al*, 2000; Senn *et al*, 2003).

Previous studies in adults have suggested baseline transcriptional signatures may be predictive of subsequent responses to vaccines (Tsang *et al*, 2014). Here we used machine learning to build models that were able to predict infant immunological (MenB-specific SBA

titres) and physiological (fever) responses following vaccination, using pre-vaccination peripheral blood gene signatures as input variables. The most significant contributor to the baseline gene signature predictive of post-vaccination fever was *APBA3*, an important gene in the function and migration of macrophages and inflammatory monocytes via the glycolytic pathway (Hara *et al*, 2011). Mouse knockouts of *Apba3* have defective macrophages, that are resistant to LPS and fail to migrate to sites of acute inflammation (Hara *et al*, 2011; Uematsu *et al*, 2016). Interestingly, 3/5 (*FUZ*, *MKS1* and *KDM7A*) of the most important contributors to the baseline gene signature predictive of MenB-specific SBA titres have also been shown to be differentially expressed in monocytes stimulated with LPS (Cepika *et al*, 2017).

This project described gene regulation following immunisation of a healthy cohort of Caucasian infants, at a single site in the United Kingdom. The generalisability of these finding to infants from different ethnic, geographical, genetic and health backgrounds is unknown. We chose to characterise the blood transcriptome following immunisations given at 4 months of age, as previous data had shown the greatest rise in MenB-specific SBA titre and the greatest reactogenicity to occur following a second dose of 4CMenB vaccine (Gossger *et al*, 2012; Snape *et al*, 2013). Therefore, we are limited to evaluating the transcriptome around this dose and do not have data related to the first dose of 4CMenB. In infants, we exclusively looked at 4CMenB given concomitantly, as this is how this vaccine has been implemented into the UK infant vaccination programme. However, we supplemented our finding by exploring gene regulation induced by this vaccine when administered alone in a mouse model. We conducted bulk sequencing of whole blood samples, which presents difficulties in deducing whether changes in gene expression reflect changes at the cellular level or changes in cell abundances. To ameliorate this, we employed a computational approach to deconvolute cellular abundances which we showed to perform well for neutrophils; however, this approach has a number of limitations such as lack of fidelity of cell reference profiles and phenotypic plasticity (Newman *et al*, 2015). Caution is advised when interpreting these deconvoluted cell proportions; e.g. while the deconvoluted neutrophil proportions correlated well with actual neutrophil counts, the use of a basis/signature matrix exclusively generated from microarray datasets (LM22 or immunoStates) resulted in a systematic underestimation of neutrophil proportions in this RNA-seq dataset. In conclusion, these findings provide evidence of a relationship between particular vaccine components and increases in circulating neutrophils and their chemotaxis and post-vaccination fever. These discoveries could have important implications in the design of future vaccines.

# Material and Methods

## Reagents and Tools table

| Reagent/Resource | Reference or Source | Identifier or Catalog Number |
|---|---|---|
| **Experimental Models** | | |
| C57BL/6J (*Mus musculus*) | Harlan, UK | N/A |
| **Chemicals, Enzymes and other reagents** | | |
| PCV13 | Pfizer | Prevenar® |
| DTaP-IPV-Hib | Sanofi Pasteur MSD | Pediacel® |
| MenC-TT | Baxter Vaccines | NeisVac-C® |
| Oral rotavirus vaccine | GSK | Rotarix |
| *Escherichia coli* lipopolysaccharide | Invivogen | tlrl-eblps |
| *Escherichia coli* lipopolysaccharide ultrapure | Invivogen | tlrl-3pelps |
| *Escherichia coli* peptidoglycan | Invivogen | tlrl-pgneb |
| Alum | Invivogen | vac-alu-250 |
| MILLIPLEX® Human cytokine/chemokine panel | Merck | HCYTOMAG-60K-23C |
| MILLIPLEX® human CVD panel 3 | Merck | HCVD3MAG-67K |
| **Software** | | |
| HISAT2 | https://daehwankimlab.github.io/hisat2/manual/ | version 2.0.5 |
| STAR | https://github.com/alexdobin/STAR | version 2.6 |
| SortMeRNA | https://bioinfo.lifl.fr/RNA/sortmerna/ | version 2.1 |
| Cibersortx | https://cibersortx.stanford.edu | version 1.0 |
| ImmQuant | http://csgi.tau.ac.il/ImmQuant/ | version 25.01.2017 |
| HISAT-genotype | https://daehwankimlab.github.io/hisat-genotype/ | version 1.0.1-beta |
| R | https://www.r-project.org | version 3.4.4 |

## Reagents and Tools table   (continued)

| Reagent/Resource | Reference or Source | Identifier or Catalog Number |
| --- | --- | --- |
| edgeR | Robinson *et al* (2010) | version 3.20.9 |
| Limma | Law *et al* (2014) | version 3.34.9 |
| Caret | http://topepo.github.io/caret/index.html | version 6.0.80 |
| XGR | http://xgr.r-forge.r-project.org | version 1.1.4 |
| Tmod | https://cran.r-project.org/web/packages/tmod/tmod.pdf | version 0.36 |
| **Other** | | |
| Illumina HiSeq4000 | Illumina | |
| iButton | iButton™ | |

## Methods and Protocols

### Clinical study design

This study was a randomised, open-label, single-centre, descriptive study (NCT02080559) conducted by the Oxford Vaccine Group, University of Oxford. One hundred and eighty-seven healthy infants aged 8–12 weeks who had not yet received their routine infant immunisations were enrolled. Infants were randomly assigned (1:1) to a control group (to receive their routine vaccines according to the UK immunisation schedule, Fig 1, i.e. PCV13 [Prevenar®, Pfizer] and DTaP-IPV-Hib [Pediacel®, Sanofi Pasteur MSD] at 2 and 4 months of age) or a test group (4CMenB [GSK] plus aforementioned routine/control vaccines at 2 and 4 months of age). Control vaccines were administered in the anterolateral left thigh, while 4CMenB was administered intramuscularly in the anterolateral right thigh. Both test and control groups also received oral rotavirus vaccine (Rotarix, GSK) at 2 and 3 months; and MenC-TT at 3 months (NeisVac-C, Baxter Vaccines; Fig 1 and Appendix Fig S1). PCV13 and DTaP-IPV-Hib vaccines contained aluminium phosphate adjuvant; MenC-TT and 4CMenB vaccines contained aluminium hydroxide adjuvant. In this study, paracetamol was not given prophylactically but post-vaccination paracetamol/ibuprofen was administrated at the parent's/guardian's discretion.

All participants were Caucasian infants (defined as having two Caucasian parents). Ethnicity is a factor that is known to influence baseline gene expression characteristics, and contemporary gene expression analysis methods frequently exclude data from different ethnic groups (to reduce data heterogeneity) from downstream analysis (Spielman *et al*, 2007). Therefore, it seemed unethical to recruit participants whose data were not likely to be included in differential gene expression analysis. Blood samples for transcriptomic and proteomic analyses were taken at 4 months of age, 7 days or fewer before a second dose of 4CMenB in the test group, then either 4 h, 24 h, 3 days or 7 days post-vaccination (Fig 1). This sub-allocation was to minimise the number of blood tests each infant received. Sub-allocation was dependent on parental availability; however, re-allocation was possible if an earlier visit was unsuccessful. Blood samples for immunogenicity were taken at 5 months of age. Written informed consent was obtained from the parent or legal guardian, after a detailed explanation of the study. A national research ethics committee (South Central - Oxford A, 14/SC/0077) approved this study.

### Serum bactericidal assays

Plasma samples collected 1 month after a second dose of 4CMenB in the test vaccine group were assessed for capsular group B meningococci SBA activity (plasma or serum can be used in this assay) (Borrow *et al*, 2005). SBA using human complement was performed at Vaccine Evaluation Unit, Public Health England, Manchester using capsular group B meningococci strain H44/76-SL (Borrow *et al*, 2005). The lower limit of quantification for the SBA assays was 2; samples without detectable SBA activity were assigned an arbitrary value of 1.

### Full blood count

The clinical haematology laboratory (Oxford University Trust Hospital NHS foundation trust) performed the full blood counts (FBC).

### C-reactive protein

The clinical biochemistry laboratory (Oxford University Trust Hospital NHS foundation trust) performed the C-reactive protein (CRP) assessment using spectrophotometry.

### Reactogenicity measure (iButton)

The body temperature of participants at the time of their 4-month immunisations was recorded by using a wireless continuous temperature monitoring system (iButton®). The iButton® is a non-invasive device for human skin temperature measurements, that has the capacity to record a temperature reading each minute over a 24 h period. In this study, the iButton® was applied to the infant's abdomen and used to record temperatures in the first 24 h post-vaccination. Fever was defined as any iButton® recording of $\geq 38°C$ within the first 24 h of vaccination. The chi-square test was used to compare the iButton® fever rates between the control vaccine and 4CMenB + control vaccine groups. Time to first fever in both groups was illustrated using Kaplan–Meier failure curves, and the log-rank test was used to compare the groups. In the transcriptomic analysis, only iButton® datasets with < 90 min of missing data were considered "complete" and included in the reactogenicity analysis (to minimise misclassification).

### Mouse immunisations

All procedures were performed in accordance with the terms of the UK Home Office Animals Act Project License. The University of

Oxford Animal Care and Ethical Review Committee approved procedures. Mice were immunised intramuscularly under general anaesthesia. Cardiac bleeds were performed under general anaesthesia followed by cervical dislocation. A mouse immunisation model was designed to recapitulate the human infant study, previously described (Sheerin et al, 2019). Groups of six 6- to 8-week old female C57BL/6 mice (Harlan, UK) were immunised intramuscularly with 1/15 of the human dose (to comply with the maximum volume of vaccine allowed per mouse outlined in the project licence) for vaccine combination comparisons—4CMenB only, control group (routine vaccines only), 4CMenB + control vaccines (test group) and phosphate-buffered saline (PBS control)—or 1/5 of the human dose for single vaccine/antigen comparisons—E. coli LPS (Invivogen, France, tlrl-eblps) in alum (Invivogen, France, vac-alu-250), ultrapure E. coli LPS (Invivogen, France, tlrl-3pelps) in alum, E. coli peptidoglycan (Invivogen, tlrl-pgneb) adsorbed on alum, combination of alum-adsorbed peptidoglycan and LPS, alum only or no immunisations at all (naïve control; Dataset EV1). Blood samples were taken 24 h after the second dose (day 22) and stored in RNAprotect Animal Blood Tubes (QIAGEN) containing RNA-stabilising reagent and incubated at room temperature for 2 h to lyse blood cells.

### Human infant RNA Sequencing
Peripheral blood (up to 1.5 ml) was collected into a reduced volume PAXgene™ RNA stabilisation reagent (ratio of blood to PAXgene equivalent to manufacturer's specifications). Total RNA was extracted using the Blood RNA Kit (PreAnalytiX, Switzerland), using the automated protocol (QIAcube instrument, QIAGEN, Germany). The ribodepleted and globin depleted fraction were selected from the total RNA using Ribo-Zero™ Gold rRNA removal Kit (Illumina, USA). RNA was converted to cDNA, second-strand cDNA synthesis incorporated dUTP. The cDNA was end-repaired, A-tailed and adapter-ligated and prior to amplification, samples underwent uridine digestion (to ensure strand-specific sequencing). The prepared libraries were size selected, multiplexed and quality-controlled before 100bp paired-end sequencing (HiSeq4000). The multiplexing blocking strategy is available with the raw sequencing data (Gene Expression Omnibus, GSE131929). Sequencing was conducted at the Wellcome Trust Centre for Human Genetics (Oxford, UK).

The sequencing data (fastq) files were aligned against the whole human (Homo sapiens) genome build GRCh38 (https://ccb.jhu.edu/software/hisat2/index.shtml), using HISAT2 (version 2.0.5) (Pertea et al, 2016). Gene counting was conducted using the HTSeq (version 0.9.1), utilising human gene annotation gtf (General Transfer Format) version GRCh38.88 (www.ensembl.org). To remove genes with very low counts across most libraries, only genes with an abundance of more than three counts per million in nine or more samples were carried forward. Genes assigned a "gene biotype" of ribosomal RNA (rRNA), sex chromosome genes, mitochondrial RNA or haemoglobin were excluded from downstream analysis. Human leucocyte antigen typing of RNA-sequencing data using HISAT-genotype (version 1.0.1-beta) was used to check correct pairing of pre- and post-vaccination samples (preprint: Kim et al, 2018).

### Human infant proteome analysis
Plasma cytokines were measured at baseline and 4 and 24 h post-vaccination, in all participants for whom plasma was available. Prior to proteome measurement, plasma samples were thawed at

room temperature and then clarified by spinning at 10,000 g for 15 min at 4°C to remove any residual platelets and debris.

Twenty-six cytokines were measured in multiplex using Luminex® technology (MILLIPLEX® Multiplex Assays, Merck, USA). The Human cytokine/chemokine panel (cat # HCYTOMAG-60K-23C) and human CVD panel 3 (cat # HCVD3MAG-67K) were used to measure the following plasma proteins: L-selectin, epidermal growth factor (EGF), transforming growth factor alpha (TGF-α), granulocyte colony-stimulating factor (G-CSF), granulocyte–macrophage colony-stimulating factor (GM-CSF), fractalkine (FKN), interferon-γ (IFNγ), GRO, interleukin-1α (IL-1α), interleukin-1β (IL-1β), interleukin-2 (IL-2), interleukin-3 (IL-3), interleukin-4 (IL-4), interleukin-5 (IL-5), interleukin-6 (IL-6), interleukin-8 (IL-8), interleukin-10 (IL-10), interleukin-13 (IL-13), interleukin-17A (IL-17A), interleukin-1 receptor antagonist (IL-1RA), IP-10/CXCL10, tumour necrosis factor alpha (TNF-α) and soluble CD-40 ligand (sCD40L).

Samples and standard curves were run in duplicate. Mean fluorescence intensity (MFI) was read on a MagPix® (Luminex Corporation, USA) instrument. MFI was converted to concentration based on standard curves via the xPonent version 4.2 software (Luminex Corporation, USA) using default settings (logistic 5P weighted curve). Average concentrations were calculated for samples run in duplicate. A two-sample Wilcoxon rank sum test was applied to compare post-vaccination plasma protein levels between the concomitant 4CMenB and control vaccine groups.

### Mouse RNA sequencing
RNA was extracted from whole blood samples using a Mouse Ribo-Pure™-Blood RNA Isolation Kit (Ambion, USA). Samples were depleted of α- and β-globin messenger RNA (mRNA) transcripts using a GLOBINclear™ Mouse/Rat Kit, (Ambion, USA). Polyadenylated mRNA transcripts were selected by oligo (dT) beads, uridine digested, converted to complementary DNA, amplified and labelled using a TotalPrepTM-96 RNA Amplification Kit (Illumina, USA). The prepared libraries were size selected for 75 bp fragments and multiplexed before paired-end sequencing using an Illumina HiSeq4000 (Illumina, USA), at the Wellcome Trust Centre for Human Genetics (Oxford, UK). Reads were aligned to the mouse (M. musculus) GRCm38 reference assembly and annotation (release 93, http://www.ensembl.org) using STAR v2.6 (Dobin et al, 2013). Reads per gene were determined simultaneously by using the STAR "-quantMode GeneCounts" option. Counts were estimated at the gene level. Lowly expressed genes (fewer than 0.5 reads per million in 2 samples) were filtered from the data.

### Differential gene expression analysis
Differential gene expression was undertaken using the R Bioconductor packages "edgeR" and "limma" (Robinson et al, 2010; McCarthy et al, 2012; R core team, 2013; Ritchie et al, 2015). RNA-sequencing data were normalised for RNA composition using trimmed mean of M-value (TMM) method (Robinson & Oshlack, 2010). Data were transformed using the limma "voom" function. A linear model was fitted to the data using the limma "lmFit" function using the empirical Bayes method to borrow information between genes (Ritchie et al, 2015). For the human study, paired analysis was conducted to compare pre- and post-vaccination samples at each of the study time points and the statistical cut-off for significance was set at false discovery rate (FDR) < 0.01. For the murine study, unpaired

analysis was conducted to compare each vaccine group with a group of PBS-immunised mice, and the statistical cut-off for significance was set at FDR < 0.01.

### Cell enumeration whole blood transcriptomic data

Cell composition of infant whole blood samples was evaluated using the CIBERSORTx method (Newman *et al*, 2019). A filtered (as described above), non-log space reads per kilobase million (RKPMs) sample gene matrix and a "signature" gene files (LM22; 22 immune cell types and immunoStates; 20 immune cell types (Vallania *et al*, 2018)) were used to deconvolute cell abundances. CIBERSORTx was run in relative mode, with B-mode batch correction with 1,000 permutations, using cell type-specific gene expression profiles for the analysis (Newman *et al*, 2019). Cell composition of mouse samples was evaluated using the ImmQuant tool (Frishberg *et al*, 2016). A filtered (as described above), $log_2$ transformed sample gene matrix and an Immunological Genome Project mouse gene expression dataset were used to infer immune cells based on signature markers (Heng *et al*, 2008). Predicted relative abundances were calculated relative to the PBS-immunised mouse samples.

### Pathway and gene set enrichment analysis

Gene set enrichment analysis was performed on differentially expressed genes (FDR < 0.01) employing XGR, using the functional categories of Gene Ontologies (GO; biological process, cellular component and molecular function) (Fang *et al*, 2016; The Gene Ontology Consortium, 2017).

### Blood transcriptional modules

Blood transcriptional module analysis was undertaken using the "tmod" R package on genes ranked by their log-ratio (LR) value; statistical testing for module expression was evaluated using the "tmodCERNOtest" function, which is a non-parametric test working on gene ranks (Yamaguchi *et al*, 2008; Weiner, 2016).

### Machine learning

To identify baseline transcriptome predictors that can discriminate between infants that develop fever following 4CMenB vaccination, we applied sequential, iterative modelling "overnight", SIMON, as described previously (Tomic *et al*, 2019). Briefly, in the first step of the SIMON analysis, all data are centred and scaled; then, the dataset is divided into training (75%) and test sets (25%). The same training and test sets are used to evaluate the performance for each of the 215 algorithms tested and to select the best performing models. To evaluate model performance and test the validity of class predictors, we implemented the two-step procedure in SIMON. The accuracy of the predictors was first tested by 10-fold cross-validation on 75% of the data from the initial dataset (training set only). The cross-validation process is repeated five times, and cumulative error rate is calculated. To prevent optimistic accuracy estimates resulting from overfitting, in the last step, each model is evaluated on the withheld test set (Kohavi, 1995). The performance of classification models was determined by calculating the area under the receiver operating characteristic curve (AUROC) for training (train AUROC) and test set (test AUROC). Out of all 215

algorithms tested, only 33 algorithms (15%) successfully built models, and only 10 models had good performance (training AUROC value above 0.7; Dataset EV13). Three of these models were overfitted, having AUROC value < 0.7 as evaluated on the test set. In the final step, SIMON calculated the contribution of each feature to the model as variable importance score (scaled to maximum value of 100), as described in the R package "caret" (Kuhn, 2017). Since variable importance score for each feature is determined by using the model information, this ultimately incorporates the correlation structure between the predictors and the importance calculation.

Similarly, a supervised machine learning algorithm was used to determine a predictive model of quantitative post-vaccination (1 month after 4CMenB 4 months immunisation) MenB SBA titres based on pre-vaccination gene expression profiles. For this purpose, we utilised a radial basis function kernel support vector regression (SVR), a support vector machine (SVM) that is appropriate for regression analysis (Smola & Schölkopf, 2004). Again, a partition was created with 75 and 25% of pre-vaccination (test group only) RNA-sequencing data in a training dataset and test dataset, respectively. Highly correlated genes ($R > 0.9$) within the training dataset were removed from both datasets, which were then scaled and centred. SVR was used to tune model parameters, using 25 bootstrapped iterations of the training dataset with $log_{10}$ MenB SBA titres as the outcome measure, utilising the R package "caret" (Kuhn, 2017). The summary metric used to select the optimal model was the root-mean-square error (RMSE). The final model from the training set was then used to assess its ability to predict $log_{10}$ MenB SBA titres in the test dataset.

## Data availability

The human study and mouse datasets generated are available in the Gene Expression Omnibus (GSE131929 and GSE132199; https://www.ncbi.nlm.nih.gov/geo/).

Expanded View for this article is available online.

### Acknowledgements

This work received funding from the European Union's Seventh Framework Programme under EC-GA no. 279185 (EUCLIDS) and Medical Research Council (MRC, UK). The research was supported by the National Institute for Health Research (NIHR) Oxford Biomedical Research Centre (BRC). The views expressed are those of the author(s) and not necessarily those of the NHS, the NIHR or the Department of Health. Funding for this study was also provided by GlaxoSmithKline Biologicals SA [NCT02080559]. GlaxoSmithKline Biologicals SA was provided the opportunity to review a preliminary version of this manuscript for factual accuracy but the authors are solely responsible for final content and interpretation. CR is a Jenner Investigator, and is supported by the Oxford NIHR Biomedical Research Centre (BRC), Vaccine theme. We thank all the children and families who participated in the study, as well as the research nurses and doctors and laboratory scientists of Oxford Vaccine Group (University of Oxford, Oxford, UK). We also thank the High-Throughput Genomics Group at the Wellcome Trust Centre for Human Genetics (funded by Wellcome Trust grant reference 090532/Z/09/Z) for the generation of the Sequencing data.

## Author contribution

DO'C, ML and AJP conceptualised the project. DO'C, MVP, HR, SK, EP, MDS and AJP designed and/or managed the clinical trial. DS, CD, MS and CSR designed the mouse model study. DO'C, MVP, DS, RED, SC-W, HH and LS conducted experimental procedures included within this report. DO'C, AT and UG completed the formal analysis. ML and AJP acquired the funding for this project. All authors reviewed and approved the manuscript.

## Conflict of interest

AJP reports grants from Okairos outside the submitted work, and AJP is Chair of UK Department of Health and Social Care's Joint Committee on Vaccination and Immunisation, and the EMA scientific advisory group on vaccines and is a member of the World Health Organisation's Strategic Advisory Group of Experts on Immunisation. MDS acts as a Chief/Principal Investigator on clinical trials funded by vaccine manufacturers including GSK group of companies, Novavax, Medimmune, MCM, Pfizer and Janssen. These studies are conducted on behalf of the University of Oxford and MDS receives no personal financial benefit. MVP is a member of the Portuguese National Immunisation Technical Advisory Group (Comissão Técnica de Vacinação da Direcção Geral de Saúde).

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
