## [Review Process File · Molecular Systems Biology]

Gene expression profiling reveals insights into infant responses to group B meningococcal vaccine

Daniel O'Connor, Marta Valente Pinto, Dylan Sheerin, Adriana Tomic, Ruth Drury, Samuel Channon-Wells, Ushma Galal, Christina Dold, Hannah Robinson, Simon Kerridge, Emma Plested, Harri Hughes, Lisa Stockdale, Manish Sadarangani, Matthew Snape, Christine Rollier, Michael Levin, and Andrew Pollard

DOI: [10.15252/msb.20209888](https://doi.org/10.15252/msb.20209888)

Corresponding author(s): Daniel O'Connor (daniel.oconnor@paediatrics.ox.ac.uk)

Review Timeline:

Submission Date:	27th Jul 20
Editorial Decision:	1st Sep 20
Revision Received:	27th Sep 20
Editorial Decision:	6th Oct 20
Revision Received:	6th Oct 20
Accepted:	8th Oct 20

Editor: Maria Polychronidou

Transaction Report:

Thank you for submitting your work to Molecular Systems Biology. We have now heard back from the three referees who agreed to evaluate your study. As you will see below, the reviewers acknowledge that the study seems to be a valuable contribution to the field. However, they raise a series of concerns, which we would ask you to address in a revision.

I think that the recommendations of the reviewers are rather clear and there is therefore no need to repeat the points listed below. Most of the reviewers' comments can be addressed by relatively minor revisions. Please let me know in case you would like to discuss in further detail any of the issues raised.

On a more editorial level, we would ask you to address the following issues.

REFEREE REPORTS

Reviewer #1:

In this study, Pollard and colleagues analyze the blood transcriptional signatures and proteomic signatures following immunization of infants with DTaP-IPV-Hib plus MenB vaccine, or with DTaP-IPV-Hib vaccines alone. The data reveal insight into the molecular mechanisms underlying post-vaccination reactogenicity and immunogenicity. Blood samples were taken at 4 months of age, then either 4 hours, 24 hours, 3 days or 7 days, as well as 28 days post-vaccination. Blood gene expression profiles were assessed by 100bp paired-end RNA-sequencing and a panel of 25 plasma protein were measured. Vaccine responses were measured using a serum bactericidal assay 28 days after vaccination. 4CMenB vaccination was associated with increased gene expression of ENTPD7, and increased concentrations of 4 plasma proteins: CRP, G-CSF, IL-1RA and IL-6. Post-vaccination fever was associated with increased expression of SELL, involved in neutrophil recruitment. Moreover, in mice, the concomitant regimen was associated with increased gene perturbation compared with 4CMenB vaccine administered alone and enhancement of pathways such as interleukin-3, -5 and GM-CSF signalling. Finally, we present transcriptomic profiles that are predictive of immunological and febrile responses following 4CMenB vaccine.

This is a very timely study that will be of broad interest to the readership of MSB. The study is very well performed and provides a valuable resource in terms of transcriptomics and proteomics data on the human immune response to MenB vaccination. The identification of gene signatures that correlate with disease severity is novel.

The authors compare signatures of DTaP-IPV-Hib plus MenB versus DTaP-IPV-Hib alone. The authors should clarify what routes of immunization were used for the vaccines, and whether Men B was administered at the same route as DTaP-IPV-Hib. Also, the authors should clarify what adjuvants were used with these vaccines.

Reviewer #2:

Summary

In this study, the authors set out to examine the transcriptional changes seen in infants after undergoing vaccination with the 4CMenB vaccine. This vaccine is generally associated with an increased rate of fever when given in combination with the routine vaccinations, which is the current practice in the UK. Transcriptional changes in infants with and without the 4CMenB vaccine across various time points were evaluated alongside changes at the proteomic level and in a mice model.

The primary conclusions are that the blood transcriptome of infants drastically changes from 4 to 24 hours post-vaccination which is associated with an increase in blood neutrophils with a gene signature being able to capture this change in the two earliest timepoints (4H and 24H).

Secondly, an increase in fever was seen at a significant level in infants who received the 4CMenB vaccine only and associated with a single gene, *SELL*, which encodes the protein L-selection. L-selection is known to mediate the adherence of lymphocytes to endothelial cells which, as stated, may increase neutrophil recruitment to the vaccination site.

The study used several methodologies to examine the effect of this vaccination on infants, primarily transcriptional analysis through RNA-sequencing in both Humans and Mouse alongside differential gene expression analysis to examine significant gene differences between groups. They also employed a machine-learning algorithm to parse the gene data and identify a gene model for

prediction of vaccine response.

To the best of my knowledge, this is the first 4CMenB vaccine study that examines the transcriptional changes in infants and not adults. This manuscript, therefore, advances the knowledge around what occurs in real-world settings when the vaccines are usually received. Overall the manuscript is well described and offers a valuable addition to the field.

Minor points

Overall some points could do with a bit of clarification.

Abstract - jumps straight into discussing the mice results without first introducing the use of mice.

Unclear on the use of time points. Mention the time points as either 4H, 24H, 3 days or 7 days and 26 days post-vaccination. However, no explanation is given on the N at each Groups time point. E.g. G1 4h vs G2 4h. Furthermore, why was blood taken from some infants at specific timepoints and not others, and how was this chosen?

G1 - 4CMenB plus routine/control vaccines, n =92

G2 routine/control vaccines only, n =89

Methods - Clinical study design

The use of control vaccines when initially introducing the test and control groups is unclear. Control vaccines in the test group are initially mentioned then later in the control group; routine vaccines are used. Is it safe to assume, from reading the rest of the manuscript, that the control vaccines is the routine vaccines? However, this could be more clearly described.

Mention the used of paracetamol/ibuprofen was administrated at the parent's/guardian's discretion. Can this affect the study outcome? Was it noted in any of the metadata?

Refers to Figure 2a here, but should it not refer to figure 1 as figure 2a is the Kepler Maier failure estimates.

Methods - Human infant RNA Sequencing and Mouse RNA Sequencing

Why the use of different approaches for the human RNA-seq data vs the mouse RNA-seq data? While the library prep and length is different, and both approaches are valid; it seems an unusual choice as the use of HISAT and Htseq for humans could easily be applied to the mouse data.

Figure 1. Study Overview - same question about the timepoints as above. How was the Or chosen?

How were the timepoints chosen?

How many at

- i) 4h
- ii) 24H
- iii) 72H / 3days
- iv) 168H or 7 days

Figure 2 - a) refers both to the non-existent study overview and the Kaplan-Meier to first fever Furthermore, the x-axis in A and C is hard to read and is blurry. Can the DPI be increased? It might just be from the merged pdf provided to review. In 2A, the legend and image also seem overly stretched and is hard to view clearly. 2C - cannot easily read the yellow writing, i.e. what is

influencing PC1/2

Peak in blood transcriptome differential gene expression 24 hours after infant vaccination

It is unclear what comparison groups were used. Both G1 and G2 were combined into one group, and then the baseline was compared to each timepoint. However, what is the baseline in this case? Was it the blood taken at 4 months before the application of the 2nd round of vaccines?

G1 + G2 Baseline vs G1+ G2 timepoints (4H, 24H, 72H, 168H) Is that correct?

Baseline gene profiles predictive of immune response and reactogenicity

Use of 54 in text with no explanation of why is it 54/92 from the entire test group. How were these 54 selected? Are these the Training set with the remaining 38 being the test set?

4CMenB induces a greater magnitude of early proinflammatory gene expression in mice when administered in combination with control immunisations

In mouse what was the comparator for the differential gene expression: the PBS control or the Naive control mentioned in methods?

Figure 3

Is pre-vaccination here referring to the blood taken before the second round of vaccines at 4 months? Alternatively, is there another blood draw at the start of the study before any injections that are not mentioned.

Figure 3

What is E-value? Is it the log fold change resulting from the comparison between G1 and G2 (test vs control) at the various time points?

Figure 5

Colour legends provided in the previous figure (4) but not in this. Is it the same colour code as the previous graph where purple is the test and green is the control group.

Figure 6

f) what data is plotted here? Transformed counts of the individuals, log fold changes or another value?

i) dotted line at the bottom of graph gene names (y-axis)

Supplementary

Two files would not open when downloaded due to incorrect encoding. These are supplementary tables 13 (CSV 18KB) and 16 (CSV 26KB).

Supplementary table 3: iButton summary statistics. For the Test group n=92, however, it says only 89 participants in this group had temperature readings. Why were 3 included without temperature readings?

Reviewer #3:

The manuscript by O'Connor et al. examines the blood transcriptome and proteome following infant immunizations with or without concomitant meningococcal 4CMenB vaccination, in order to better understand the molecular mechanisms underlying post-vaccination reactogenicity and immunogenicity. The study is a unique opportunity to understand infant responses to the new 4CMenB vaccine within the regular immunization schedule. Blood samples were collected from infants receiving control immunizations (PCV13 and DTaP-IPV-Hib) with or without 4CMenB at 2 and 4 months of age with five time points: 4 hours, 24 hours, 3 days, 7 days and 28 days post-vaccination after the 4 month visit. Blood gene expression was assessed by RNA-seq and a panel of 25 plasma proteins were measured. Serum bactericidal titers were determined with the 28-day sera. In addition, mouse immunizations with the infant's regimen, as well as the 4CMenB vaccine alone were conducted. Mouse immunizations with *E. coli* LPS and *E. coli* peptidoglycan (PG) alone or in combination were also done in order to examine the pyrogenicity profile evoked by 4CMenB vaccine in which LOS and PG are present in the OMV component.

Comments:

Overall, this is a very high-quality study. The authors proposed transcriptomic profiles that are predictive of immunological and febrile responses following 4CMenB vaccine. A machine learning approach was also performed to identify baseline transcriptome predictors that can discriminate between infants that develop fever following 4CMenB vaccination. While a large set of transcriptome and proteome data were collected and different bioinformatic analyses were conducted, the conclusions in relation to reactogenicity vs immunogenicity and significance need better clarity. How will the transcriptomic profiles that are predictive of immunological and febrile responses following 4CMenB impact vaccine design or clinical practice? While limitations of the computational approach are described in the discussion, it is difficult to know if the methodologies applied are adequate and if the conclusions are justified based on these methodologies. Can the authors provide additional evidence for their validity?

Specific comments: Page 2, Introduction: The authors stated that "While susceptibility to invasive meningococcal disease is not completely understood, an inverse relationship is seen with the prevalence of complement-dependent serum bactericidal antibody (SBA) titers, and the levels of SBA correlate with post-immunization protection at the population level." Clarify what inverse relationship is being referred to.

While the goal is to understand the underlying mechanisms vaccine reactogenicity and immunogenicity, what was the rationale provided in how the sampling time points were chosen. Can one assume that shorter time points may reflect reactogenicity, while the comparison between gene expression profiles of later time points reveals immunogenicity?

When 4CMenB is given concomitantly with other routine infant immunizations, fever is a common adverse reaction (>60%) and is indistinguishable from coincidental infection, resulting in unnecessary medical visits, invasive procedures and laboratory investigations. While the blood RNA seq data appears to help in discerning the etiology of acute febrile illness in children, how will the transcriptome studies or a subset potentially benefit or affect the clinical practice treating post vaccine febrile infants.

Page 11, "CD177, a neutrophil specific gene involved in neutrophil activation, was the leading contributor to principal component 2", better explain why CD177 is the leading contributor to PC2 biologically? What are the significances of the other two genes plotted in the contribution plot?

Page 12, The neutrophil fraction determined from the transcriptome data (CIBERSORT) mirrored neutrophil counts in the full blood count, and the two measures were highly correlated. Is the transcriptional perturbation simply associated with an increase in neutrophils post-vaccination?

Page 13, Supplemental Figures 5-9, the Venn diagram and the gene ontologies indicated that the control group has its own unique transcriptional changes (e.g. Suppl Fig. 5 showing 543 genes at 24-hr). Since the test group also received the two control vaccines, PCV13 and DTaP-IPV-Hib, in

addition to 4CMenB, do these data suggest that 4CMenB can interfere with responses to those two vaccines?

Figure 6, a SVR model was proposed to predict post-vaccination MenB SBA titers and the top 5 genes contributing to this model were shown. There are no descriptions of the gene functions and why these genes contribute to the predictive values of SBA titers.

Figure 7, the mouse immunization studies, while correlated, do not "validate" the observations of infant immunization. The top differentially expressed genes identified in the mouse study are different from those from the infant's study (Figures 3 and 4).

Minor comments:

Page 4, the schematic for vaccine trial is in Figure 1, not Figure 2a.

There are no files labeled as supplementary tables 1-13 that are referenced throughout the manuscript.

Figure 2b, the contribution plot does not have scale and the light yellow color of labeling is very difficult to read.

Figure 2C, the x-axis labeling needs to be corrected.

Reviewer #1:

In this study, Pollard and colleagues analyze the blood transcriptional signatures and proteomic signatures following immunization of infants with DTaP-IPV-Hib plus MenB vaccine, or with DTaP-IPV-Hib vaccines alone. The data reveal insight into the molecular mechanisms underlying post-vaccination reactogenicity and immunogenicity. Blood samples were taken at 4 months of age, then either 4 hours, 24 hours, 3 days or 7 days, as well as 28 days post-vaccination. Blood gene expression profiles were assessed by 100bp paired-end RNA-sequencing and a panel of 25 plasma protein were measured. Vaccine responses were measured using a serum bactericidal assay 28 days after vaccination. 4CMenB vaccination was associated with increased gene expression of ENTPD7, and increased concentrations of 4 plasma proteins: CRP, G-CSF, IL-1RA and IL-6. Post-vaccination fever was associated with increased expression of SELL, involved in neutrophil recruitment. Moreover, in mice, the concomitant regimen was associated with increased gene perturbation compared with 4CMenB vaccine administered alone and enhancement of pathways such as interleukin-3, -5 and GM-CSF signalling. Finally, we present transcriptomic profiles that are predictive of immunological and febrile responses following 4CMenB vaccine.

This is a very timely study that will be of broad interest to the readership of MSB. The study is very well performed and provides a valuable resource in terms of transcriptomics and proteomics data on the human immune response to MenB vaccination. The identification of gene signatures that correlate with disease severity is novel.

The authors compare signatures of DTaP-IPV-Hib plus MenB versus DTaP-IPV-Hib alone. The authors should clarify what routes of immunization were used for the vaccines, and whether Men B was administered at the same route as DTaP-IPV-Hib. Also, the authors should clarify what adjuvants were used with these vaccines.

- [Response] — We thank the reviewer for their positive and constructive comments. We have inserted the comments below in the main text, in order to clarify the routes of administration and the adjuvants used.
- [Inserted, page 6] — “Control vaccines were administered in the anterolateral left thigh, while 4CMenB was administered intramuscularly in the anterolateral right thigh.”
- [Inserted, page 6] — “PCV13 and DTaP-IPV-Hib vaccines contained aluminium phosphate adjuvant; MenC-TT and 4CMenB vaccines contained aluminium hydroxide adjuvant.”

Reviewer #2:

Summary

In this study, the authors set out to examine the transcriptional changes seen in infants after undergoing vaccination with the 4CMenB vaccine. This vaccine is generally associated with an increased rate of fever when given in combination with the routine vaccinations, which is the current practice in the UK. Transcriptional changes in infants with and without the 4CMenB vaccine across various time points were evaluated alongside changes at the proteomic level and in a mice model.

The primary conclusions are that the blood transcriptome of infants drastically changes from 4 to 24 hours post-vaccination which is associated with an increase in blood neutrophils with a gene signature being able to capture this change in the two earliest timepoints (4H and 24H).

Secondly, an increase in fever was seen at a significant level in infants who received the 4CMenB vaccine only and associated with a single gene, *SELL*, which encodes the protein L-selection. L-selection is known to mediate the adherence of lymphocytes to endothelial cells which, as stated, may increase neutrophil recruitment to the vaccination site.

The study used several methodologies to examine the effect of this vaccination on infants, primarily transcriptional analysis through RNA-sequencing in both Humans and Mouse alongside differential gene expression analysis to examine significant gene differences between groups. They also employed a machine-learning algorithm to parse the gene data and identify a gene model for prediction of vaccine response.

To the best of my knowledge, this is the first 4CMenB vaccine study that examines the transcriptional changes in infants and not adults. This manuscript, therefore, advances the knowledge around what occurs in real-world settings when the vaccines are usually received. Overall the manuscript is well described and offers a valuable addition to the field.

Minor points

Overall some points could do with a bit of clarification.

Abstract - jumps straight into discussing the mice results without first introducing the use of mice.

- [Response] — We thank the reviewer for their helpful feedback. The abstract format for MSB is very brief (175 words) but we have modified the introduction to the mouse data in the abstract as below.
- [Edited, page 1] — “A murine model dissecting the vaccine components, found the concomitant regimen to be associated with increased gene perturbation compared with 4CMenB vaccine alone with enhancement of pathways such as interleukin-3,-5 and GM-CSF signalling.”

Unclear on the use of time points. Mention the time points as either 4H, 24H, 3 days or 7 days and 26 days post-vaccination. However, no explanation is given on the N at each Groups time point. E.g. G1 4h vs G2 4h.

Furthermore, why was blood taken from some infants at specific timepoints and not others, and how was this chosen?

- [Response] — In order to minimise the volume of blood and number of blood tests each infant had — as they are small infants — participants were sub-allocated into the four time points described. Sub-allocation was dependent on parental availability; however, re-allocation to a different sub-allocation was possible if an earlier visit was unsuccessful. We have edited the methods section as below to clarify this rationale for this design.
- [Edited, page 6] — “Blood samples for transcriptomic and proteomic analyses were taken at 4 months of age, 7 days or fewer before a second dose of 4CMenB in the test group, then either 4 hours, 24 hours, 3 days or 7 days post-vaccination (Figure 2a). This sub-allocation was to minimise the number of blood tests each infant received. Sub-allocation was dependent on parental availability; however, re-allocation was possible if an earlier visit was unsuccessful.”

G1 - 4CMenB plus routine/control vaccines, n =92

G2 routine/control vaccines only, n =89

Methods - Clinical study design

The use of control vaccines when initially introducing the test and control groups is unclear. Control vaccines in the test group are initially mentioned then later in the control group; routine vaccines are used. Is it safe to assume, from reading the rest of the manuscript, that the control vaccines is the routine vaccines? However, this could be more clearly described.

- [Response] — Thank you for the constructive comment — your assumption is correct, we have edited the methods section to clarify this important detail.
- [Edited, page 5/6] — “Infants were randomly assigned (1:1) to a control group (to receive their routine vaccines according to the UK immunisation schedule, Figure 1, i.e. PCV13 [Prevenar®, Pfizer] and DTaP-IPV-Hib [Pediace®], Sanofi Pasteur MSD] at 2 and 4 months of age) or a test group (4CMenB [GSK Vaccines] plus aforementioned routine/control vaccines at 2 and 4 months of age).”

Mention the used of paracetamol/ibuprofen was administrated at the parent's/guardian's discretion. Can this affect the study outcome? Was it noted in any of the metadata?

- [Response] — These metadata were collected; however, as paracetamol/ibuprofen was administrated at the parent's/guardian's discretion it was highly heterogeneous (timing and number of doses), prohibiting formal statistical analysis, as stratification would result in prohibitively small subgroups.

Refers to Figure 2a here, but should it not refer to figure 1 as figure 2a is the Kepler Maier failure estimates.

- [Response] — This has been corrected to Figure 1

Methods - Human infant RNA Sequencing and Mouse RNA Sequencing

Why the use of different approaches for the human RNA-seq data vs the mouse RNA-seq data? While the library prep and length is different, and both approaches are valid; it seems an unusual choice as the use of HISAT and Htseq for humans could easily be applied to the mouse data.

- [Response] — As you have highlighted both approaches are valid. This difference in approach was simply the result of analyst's preference and their established analytical pipelines. DO'C analysed the human data and DS analysed the mouse data. Moreover, on another dataset we directly compared STAR and HISAT alignment, and found consistent results <https://www.biorxiv.org/content/10.1101/843789v1>.

Figure 1. Study Overview - same question about the timepoints as above. How was the Or chosen?

How were the timepoints chosen?

How many at

i) 4h

ii) 24H

iii) 72H / 3days

iv) 168H or 7 days

- [Response] — The number of individual that had each of the time points is in Appendix Figure 1. To make this clear we have also included there numbers in the main text.
- [Inserted, page 12] — “All infants had a baseline blood sample; and 28, 31, 30 and 36 infants had a blood sample at 4 hours, 24 hours, 3 days and 7 days, respectively.”

Figure 2 - a) refers both to the non-existent study overview and the Kaplan-Meier to first fever

Furthermore, the x-axis In A and C is hard to read and is blurry. Can the DPI be increased? It might just be from the merged pdf provided to review. In 2A, the legend and image also seem overly stretched and is hard to view clearly. 2C - cannot easily read the yellow writing, i.e. what is influencing PC1/2

- [Response] — This has been corrected to remove the erroneous reference to a Study overview. The resolution of the image seems to be lost during the merging process. We have now included a high resolution image. Figure 2A has been corrected and the we have increased the font of the genes depicted in figure 2B, contributing to PC1/2

Peak in blood transcriptome differential gene expression 24 hours after infant vaccination

It is unclear what comparison groups were used. Both G1 and G2 were combined into one group, and then the baseline was compared to each timepoint. However, what is the baseline in this case? Was it the blood taken at 4 months before the application of the 2nd round of vaccines?

- [Response] — It is indeed combining test and control group infants with the baseline taken before their 4 month vaccinations, with have edited this section to make this clearer.

- [Edited, page 13] — “In this study, the greatest number of differentially expressed genes (DEGs, FDR <0.01) compared with baseline (pre-vaccination at 4 months), when the two vaccine groups (test and control) were combined, was seen 24 hours post-vaccination (DEGs = 5553) (Figure 3).”

G1 + G2 Baseline vs G1+ G2 timepoints (4H, 24H, 72H, 168H) Is that correct?

- [Response] — correct — test and control infants were combined in the initial analysis

Baseline gene profiles predictive of immune response and reactogenicity

Use of 54 in text with no explanation of why is it 54/92 from the entire test group. How were these 54 selected? Are these the Training set with the remaining 38 being the test set?

- [Response] — The reduction from 92 randomised into the test group and the 54 included in this analysis are twofold 1) difficulties in getting to successive, successful blood samples from small infants enrolled in vaccine trial (very limited number of venepuncture attempts permitted, in this setting). We only RNA-sequenced samples from individuals with paired samples. 2) We only included data from infants where the continuous temperature measurement device was successfully attached for 24 hours, with < 90 minutes of missing data.

4CMenB induces a greater magnitude of early proinflammatory gene expression in mice when administered in combination with control immunisations

In mouse what was the comparator for the differential gene expression: the PBS control or the Naive control mentioned in methods?

- [Response] — This is compared with the PBS control — thank you for highlighting this — we have now clarified as below
- [Edited, page 17] — “Mice in the test group exhibited greater perturbation of the transcriptome (695 DEGs vs PBS controls), compared with those receiving the 4CMenB vaccine alone (268 DEGs vs PBS controls) or control vaccines alone (1 DEG vs PBS controls)) (Appendix Figure S13a).”

Figure 3

Is pre-vaccination here referring to the blood taken before the second round of vaccines at 4 months?

Alternatively, is there another blood draw at the start of the study before any injections that are not mentioned.

- [Response] — This is indeed prior to vaccination at 4 months of age, we have now clarified this in the legend
- [Edited, page 25; Figure 3 legend] — “*Volcano plot highlighting differentially expressed genes (DEGs, false-discovery rate [FDR] <0.01; red upregulated and blue downregulated) at each study time point versus pre-vaccination (4 months of age)*”

Figure 3

What is E-value? Is it the log fold change resulting from the comparison between G1 and G2 (test vs control) at the various time points?

- [Response] — The expression E values, are the gene expression values derived from the voom-limma workflow, which are related to the genes log(count per million). We have included the important reference to how this was calculated in the figure legend.
- [Edited page 25, figure 4 legend] — “The expression E value, are the gene expression value derived from the voom-limma workflow (Law et al, 2014).”

Figure 5

Colour legends provided in the previous figure (4) but not in this. Is it the same colour code as the previous graph where purple is the test and green is the control group.

- The colour scheme is indeed consistent, we did not include this legend for Figure 5, as the x-axis also details the groups — we feel this is better aesthetically.

Figure 6

f) what data is plotted here? Transformed counts of the individuals, log fold changes or another value?

i) dotted line at the bottom of graph gene names (y-axis)

- [Response] — f) This is also an expression E values, as previously described. We have not correctly labelled this.
- [Response] — i) depicts the most important genes (top 5) in order of important in contribution to the SVR model.

Supplementary

Two files would not open when downloaded due to incorrect encoding. These are supplementary tables 13 (CSV 18KB) and 16 (CSV 26KB).

- [Response] — The formatting of these tables has now been resolved

Supplementary table 3: iButton summary statistics. For the Test group n=92, however, it says only 89 participants in this group had temperature readings. Why were 3 included without temperature readings?

- [Response] — Correct, no iButton readings were taken for these infants.

Reviewer #3:

The manuscript by O'Connor et al. examines the blood transcriptome and proteome following infant immunizations with or without concomitant meningococcal 4CMenB vaccination, in order to better understand the molecular mechanisms underlying post-vaccination reactogenicity and immunogenicity. The study is a unique opportunity to understand infant responses to the new 4CMenB vaccine within the regular immunization schedule. Blood samples were collected from infants receiving control immunizations (PCV13 and DTaP-IPV-Hib) with or without 4CMenB at 2 and 4 months of age with five time points: 4 hours, 24 hours, 3 days, 7 days and 28 days post-vaccination after the 4 month visit. Blood gene expression was assessed by RNA-seq and a panel of 25 plasma proteins were measured. Serum bactericidal titers was determined with the 28-day sera. In addition, mouse immunizations with the infant's regimen, as well as the 4CMenB vaccine alone were conducted. Mouse immunizations with *E. coli* LPS and *E. coli* peptidoglycan (PG) alone or in combination were also done in order to examine the pyrogenicity profile evoked by 4CMenB vaccine in which LOS and PG are present in the OMV component.

Comments:

Overall, this is a very high-quality study. The authors proposed transcriptomic profiles that are predictive of immunological and febrile responses following 4CMenB vaccine. A machine learning approach was also performed to identify baseline transcriptome predictors that can discriminate between infants that develop fever following 4CMenB vaccination. While a large set of transcriptome and proteome data were collected and different bioinformatic analyses were conducted, the conclusions in relation to reactogenicity vs immunogenicity and significance need better clarity. How will the transcriptomic profiles that are predictive of immunological and febrile responses following 4CMenB impact vaccine design or clinical practice? While limitations of the computational approach are described in the discussion, it is difficult to know if the methodologies applied are adequate and if the conclusions are justified based on these methodologies. Can the authors provide additional evidence for their validity?

- [Response] — We thank the reviewer for the thoughtful and positive consideration of this manuscript. We believe this work provides of framework for both understanding and predicting vaccine immunogenicity and reactogenicity, which may have important implications in future vaccination strategies. An important consideration is the generalisability of these findings, for infants in wider context of vaccination (i.e. different ethnic, geographical, genetic and health backgrounds which merits further investigation). This rich dataset of broad transcriptomic and proteomic data will certainly be a useful resource to aid in the design of further research in other populations. Moreover, we showed CRP and neutrophil counts are often raised post-immunisation, as these are commonly used as diagnostic markers in suspected sepsis this is clinically relevant information. Therefore, it is conceivable that these data could contribute to the clinical evaluation of febrile events occurring post-immunisation. However, a direct comparison of transcriptomic/proteomic data from different aetiologies of acute febrile illnesses is likely to be needed to develop a transcriptomic/proteomic signature that would be robust enough to influence clinical decision making.

Specific comments: Page 2, Introduction: The authors stated that "While susceptibility to invasive meningococcal disease is not completely understood, an inverse relationship is seen with the prevalence of complement-dependent serum bactericidal antibody (SBA) titers, and the levels of SBA correlate with post-immunization protection at the population level." Clarify what inverse relationship is being referred to.

- [Response] — We thank the reviewer for this comment, we have edited this sentence to clarify to what the inverse relationship relates.
- [Edited, page 2] — "While susceptibility to invasive meningococcal disease is not completely understood, an inverse relationship is seen with the prevalence of complement-dependent serum bactericidal antibody (SBA) titres and the incidence of meningococcal meningitis (Goldschneider et al, 1969). Moreover, the levels of SBA correlate with post-immunisation protection from meningococcal disease at the population level (Andrews et al, 2003)."

While the goal is to understand the underlying mechanisms vaccine reactogenicity and immunogenicity, what was the rationale provided in how the sampling time points were chosen. Can one assume that shorter time points may reflect reactogenicity, while the comparison between gene expression profiles of later time points reveals immunogenicity?

- [Response] — That is the rationale we used — we wished to capture as much of underlying mechanisms involved in reactogenicity and immunogenicity, while minimising the number of blood samples taken from these small infants.

When 4CMenB is given concomitantly with other routine infant immunizations, fever is a common adverse reaction (>60%) and is indistinguishable from coincidental infection, resulting in unnecessary medical visits, invasive procedures and laboratory investigations. While the blood RNA seq data appears to help in discerning the etiology of acute febrile illness in children, how will the transcriptome studies or a subset potentially benefit or affect the clinical practice treating post vaccine febrile infants.

- [Response] — We show here that CRP and neutrophil counts — commonly used as diagnostic biomarkers in suspected sepsis — are often raised post-immunisation. So, we show that these traditional biomarkers may not be helpful in clinical evaluation of febrile events occurring post-immunisation. However, it seem likely that a direct comparison of transcriptomic/proteomic data from different aetiologies of acute febrile illnesses is really needed in order to develop a transcriptomic/proteomic signature that would be robust enough to be utilised in this clinical context.

Page 11, "CD177, a neutrophil specific gene involved in neutrophil activation, was the leading contributor to principal component 2", better explain why CD177 is the leading contributor to PC2 biologically? What are the significances of the other two genes plotted in the contribution plot?

- [Response] — This plot shows the contribution values of the genes that contribute most to the 1st and 2nd principal components. Another way to think of these genes (possible more intuitive) is that they are most correlated with these principal components. We have edited this section to explain this more clearly.
- [Edited, page 13] — “Analysis of the genes with the greatest contribution to this clustering revealed that CD177, a neutrophil specific gene involved in neutrophil activation, was the leading contributor to (i.e. correlate with) principal component 2 (Figure 2b).”

Page 12, The neutrophil fraction determined from the transcriptome data (CIBERSORT) mirrored neutrophil counts in the full blood count, and the two measures were highly correlated. Is the transcriptional perturbation simply associated with an increase in neutrophils post-vaccination?

- [Response] — The neutrophil abundance is important in the early (4hr and 24hr) global transcriptional perturbation but even when this is adjusted for, there remains 161 DEGs at 4hrs and 3228 DEGs at 24hrs. The latter timepoints 3 days and 7 days are not correlated with neutrophil counts. This is described in the results section as below and in Appendix Figure S4.
- [Cross reference, page 13/14] — “Of note, adjusting for neutrophil abundance reduced the number of DEGs at 4 hours (719 vs. 161 DEGs) and 24 hours (5553 vs. 3228 DEGs) post-vaccination (Appendix Figure S4). DEGs were also observed at the later time points, although fewer than at the early time points, 3 days (DEGs = 159) and 7 days (DEGs =6) (Figure 3c–d).”

Page 13, Supplemental Figures 5-9, the Venn diagram and the gene ontologies indicated that the control group has its own unique transcriptional changes (e.g. Suppl Fig. 5 showing 543 genes at 24-hr). Since the test group also received the two control vaccines, PCV13 and DTaP-IPV-Hib, in addition to 4CMenB, do these data suggest that 4CMenB can interfere with responses to those two vaccines?

- [Response] — While we cannot rule out interference, we think this observation is more likely statistical/stochastic, as there is a probability of calling a given gene differentially expression (FDR) in each group — so don't expect entire overlap even in repeated experiments. Also, the group sizes are not identical — so the statistical power will differ. We have shown agreement plots (Appendix Figure S6) to evaluate this further, and we do not see evidence for interference in these analyses.

Figure 6, a SVR model was proposed to predict post-vaccination MenB SBA titers and the top 5 genes contributing to this model were shown. There are no descriptions of the gene functions and why these genes contribute to the predictive values of SBA titers.

- [Response] — The importance measure of these statistical models relates to the performance of the developed model to predict the outcome (SBA titres), when these features are dropped. Therefore, a

mechanistic/causal link to the outcome is not always apparent. Nevertheless, 3/5 of the most important genes in the model predictive SBA titres have been shown to be differentially expressed in human monocytes stimulated with lipopolysaccharide. This is interesting particular given the results of predictive model of fever, also heavily implicated monocyte/macrophages. Therefore, we have also now briefly included this in the discussion section.

- [Edited, page 23] — The most significant contributor to the baseline gene signature predictive of post-vaccination fever was *APBA3*, an important gene in the function and migration of macrophages and inflammatory monocytes via the glycolytic pathway (Hara et al, 2011). Mouse knockouts of *Apba3* have defective macrophages, that are resistant to LPS and fail to migrate to sites of acute inflammation (Hara et al., 2011; Uematsu et al, 2016). Interestingly, 3/5 (*FUZ*, *MKS1* and *KDM7A*) of the most important contributors to the baseline gene signature predictive of MenB-specific SBA titres have also been shown to be differentially expressed in monocytes stimulated with LPS (Cepika et al, 2017).

Figure 7, the mouse immunization studies, while correlated, do not "validate" the observations of infant immunization. The top differentially expressed genes identified in the mouse study are different from those from the infant's study (Figures 3 and 4).

- [Response] — We aimed to validate the infant data using a murine model. We found the pattern of differential expression between the species to be consistent/correlated, for example the *PGLYRP1* is amongst the most DEGs in test vs control group comparisons (Figure 4 and Figure 7) — in both species. We have commented in discussion section the consistencies between the species, but we also note the limitation of this model — for example the considerable differences in the peripheral blood neutrophil abundance/composition. We have edited the discussion section slightly to reflect the reviewers' concern about the use of the term "validate" — see below.
- [Edited, page 20–22] — "We aimed to validate the infant gene expression data using a murine model. We immunised mice with control immunisations, with or without the 4CMenB test vaccine, and included a 4CMenB only group. Firstly, we showed a greater magnitude of change in the blood gene profiles of mice immunised with 4CMenB vaccine than those immunised with control immunisations only. Moreover, cell deconvolution analysis showed mice immunised with 4CMenB also had an increase in their neutrophil gene signature, which was not observed in mice immunised with control immunisations only. Notably, the peptidoglycan receptor *Pglyrp1* and the triggering receptor expressed on myeloid cells *Trem1* were amongst the most upregulated genes in mice immunised with 4CMenB. *PGLYRP1* has been characterised as a ligand for the TREM-1 receptor (Pelham et al, 2014). TREM-1 is upregulated on neutrophils and monocytes during bacterial infection and, when ligated, acts synergistically with LPS to amplify the pro-inflammatory response (Bouchon et al, 2001). In order for *PGLYRP1* to stimulate TREM-1 it must be cross-linked with peptidoglycan, whereby it induces cytokine production in neutrophils and macrophages (Read et al, 2015). In the infant study we found *PGLYRP1* and *TREM1* to be significantly upregulated 4 and 24 hours post-vaccination in both vaccine groups. Of note, 4 hours after vaccination, *PGLYRP1* was one of the most differentially regulated genes in the concomitant 4CMenB group compared with the control vaccine group. Although *PGLYRP1* expression is restricted to neutrophils (in blood cells), correcting for neutrophil counts did not remove the difference in expression seen between the concomitant 4CMenB group and the control group (Liu et al, 2000). *PGLYRP1* expression is upregulated by stimulation with peptidoglycan (a cell wall component of both Gram-positive and Gram-negative bacteria) (Uehara et al, 2005). Peptidoglycan fragments are known to covalently attach to some of the purified pneumococcal polysaccharides within PCV13 (Sørensen et al, 1990). Moreover, while the peptidoglycan content of detergent extracted OMVs (component of 4CMenB) has not been described, naturally occurring OMVs do contain peptidoglycan (van der Pol et al., 2015). One explanation for the differences seen in *PGLYRP1* regulation between those who received concomitant 4CMenB and those who received control vaccines alone is a dose dependent difference in the amount of peptidoglycan delivered in the study vaccine regimens. Here, in mice, we showed that peptidoglycan alone could evoke a rise in temperature, and presented data suggesting an additive pyrogenic effect of administration of LPS plus peptidoglycan. These data suggest that the increased reactogenicity of 4CMenB when given concomitantly with other immunisations, compared with 4CMenB alone (Gossger et al., 2012), may be attributable to the additive effects of pyrogenic components of these vaccines. A noteworthy feature of infant 4CMenB vaccination is that when administered alone, fever rates are similar to those seen after other infant immunisation regimens (PCV7 and DTaP-IPV-Hib) but when given concomitantly with other childhood vaccines, fever rates are increased (Gossger et al., 2012). This additional reactogenicity has been attributed to the OMV component, as concomitant vaccination with the recombinant 4CMenB proteins and other routine vaccines (DTaP-HBV-IPV/Hib and PCV7) displays lower reactogenicity than concomitant OMV containing 4CMenB (Esposito et al., 2014). It has been suggested that attenuating LPS, by genetic modification (e.g. *lpxL1*), may ameliorate the reactogenicity of the OMV containing vaccines (Dowling et al., 2016). However, *IL1 β* (a pro-inflammatory cytokine and endogenous pyrogen) has been found to be upregulated similarly in mice immunised with native and *lpxL1* mutant OMV-containing vaccines, suggesting that non-endotoxin pyrogens, such as peptidoglycan, may also contribute to OMV reactogenicity (Sheerin et al., 2019). Moreover, in mice, the concomitant regimen was associated with increased gene perturbation compared with 4CMenB vaccine administered alone and enhancement of pathways such as interleukin-3, -5 and

GM-CSF signalling. These findings highlight the importance of fully characterising the pyrogen content of vaccines and understanding how these may interact when administered concomitantly. Despite substantial differences in peripheral blood neutrophil abundance, between mice and human infants (female 8-week C57BL/6 mice, neutrophil count $\sim 0.25 \times 10^9/L$; 2–6-month human infants, neutrophil count $1\text{--}8.5 \times 10^9/L$), we observed similar trends in gene expression profiles between the species (The Jackson Laboratory, 2007; Virgo). While the mouse model has clear limitations, these data support the utility of this model in exploring experimental procedures not amenable in human subjects.

Minor comments:

Page 4, the schematic for vaccine trial is in Figure 1, not Figure 2a.

[Response] — This has now been corrected

There are no files labeled as supplementary tables 1-13 that are referenced throughout the manuscript.

[Response] — This has now been corrected with all Datasets EV[X] cross referenced in the manuscript attached.

Figure 2b, the contribution plot does not have scale and the light yellow color of labeling is very difficult to read.

[Response] — The font has been increased for the labels in Figure 2b

Figure 2C, the x-axis labeling needs to be corrected.

[Response] — Figure 2C x-axis labelled has been checked

Thank you for sending us your revised manuscript. We think that the performed revisions have satisfactorily addressed the issues raised by the reviewers. I am therefore glad to inform you that we can soon formally accept the study for publication, pending some minor issues listed below:

2nd Authors' Response to Reviewers**6th Oct 2020**

The Authors have made the requested editorial changes.

Accepted**8th Oct 2020**

Thank you again for sending us your revised manuscript. We are now satisfied with the modifications made and I am pleased to inform you that your paper has been accepted for publication.

Corresponding Author Name: Daniel O'Connor
 Journal Submitted to: Molecular Systems Biology
 Manuscript Number: MSB-20-9888